# Reconstructing Earth's atmospheric oxygenation history using machine learning

Guoxiong Chen [1], Qiuming Cheng [1,2,3] ✉, Timothy W. Lyons[4], Jun Shen [1], Frits Agterberg[5], Ning Huang[1] & Molei Zhao[2]

Reconstructing historical atmospheric oxygen ($O_2$) levels at finer temporal resolution is a top priority for exploring the evolution of life on Earth. This goal, however, is challenged by gaps in traditionally employed sediment-hosted geochemical proxy data. Here, we propose an independent strategy—machine learning with global mafic igneous geochemistry big data to explore atmospheric oxygenation over the last 4.0 billion years. We observe an overall two-step rise of atmospheric $O_2$ similar to the published curves derived from independent sediment-hosted paleo-oxybarometers but with a more detailed fabric of $O_2$ fluctuations superimposed. These additional, shorter-term fluctuations are also consistent with previous but less well-established suggestions of $O_2$ variability. We conclude from this agreement that Earth's oxygenated atmosphere may therefore be at least partly a natural consequence of mantle cooling and specifically that evolving mantle melts collectively have helped modulate the balance of early $O_2$ sources and sinks.

An important goal for geoscientists is to decode the long-term evolution of Earth geosphere, atmosphere, and biosphere, as well as their interactions (Fig. 1). However, deciphering the evolution of early Earth is challenged by the incomplete preservation of the geological records[1]. As such, resolving various geoscience conundrums relies on deductive or inductive reasoning (i.e., a knowledge-driven model) based on (local) observations and experiments. However, given the vast and ever-expanding global geo-databases available (e.g., EarthChem, GEOROC, PetDB, SGPP, etc.), we are now in a unique position to use big data analytical techniques to answer fundamental geoscience questions via an abductive model[2,3]. This opportunity lies with data mining of increasingly large and intensive geo-datasets and from computationally intensive simulations[4,5]. In other words, although geoscientists can use logical reasoning to analyze and interpret geo-data, abductive reasoning based on machine learning with big data can facilitate data-driven discovery of previously hidden correlations or patterns that escape human intuition[3]. At the core of artificial intelligence and data science, machine learning methods guided by training experiences can

be are exceptionally good at processing big data and making decisions and predictions[6,7]. This advantage elevates their relevance in geoscience research[4,8,9], where high-resolution measurements and observations over long periods of geological time have produced vast numbers of diverse and complex data bases.

Atmospheric oxygenation is critical to the development of Earth's habitability. Oxygen levels, as shown in Fig. 1, were very low ($<10^{-7}$–$10^{-5}$ of the present atmospheric level [PAL]) for the first half of Earth history and then rose to something like $10^{-3}$–$10^{-1}$ PAL during the Great Oxidation Event (GOE)[10–12] around 2.5 to 2.0 billion years ago (Ga), although the magnitude, fabric, and timing of this jump are not well known—with ongoing debate[13,14]. Afterward, it remained at intermediate but still relatively low levels ($10^{-4}$–$10^{-1}$ PAL) for the "boring" billion years during the mid-Proterozoic[15] and then rose a second time during the Neoproterozoic Oxidation Event (NOE), achieving current oxygen levels (~1 PAL) during the Phanerozoic[12]. However, despite agreement about the first-order pattern of atmospheric oxygenation, the controls on this evolution and the second-order variations remain poorly known and are subject to debate.

[1]State Key Laboratory of Geological Processes and Mineral Resources, China University of Geosciences, Wuhan 430074, China. [2]State Key Laboratory of Geological Processes and Mineral Resources, China University of Geosciences, Beijing 10083, China. [3]School of Earth Science and Engineering, Sun Yat-sen University, Zhuhai 51900, China. [4]Department of Earth and Planetary Sciences, University of California, Riverside, CA 92521, USA. [5]Geological Survey of Canada, 601 Booth Street, Ottawa, Ontario K1A 0E8, Canada. ✉e-mail: qiuming.cheng@iugs.org

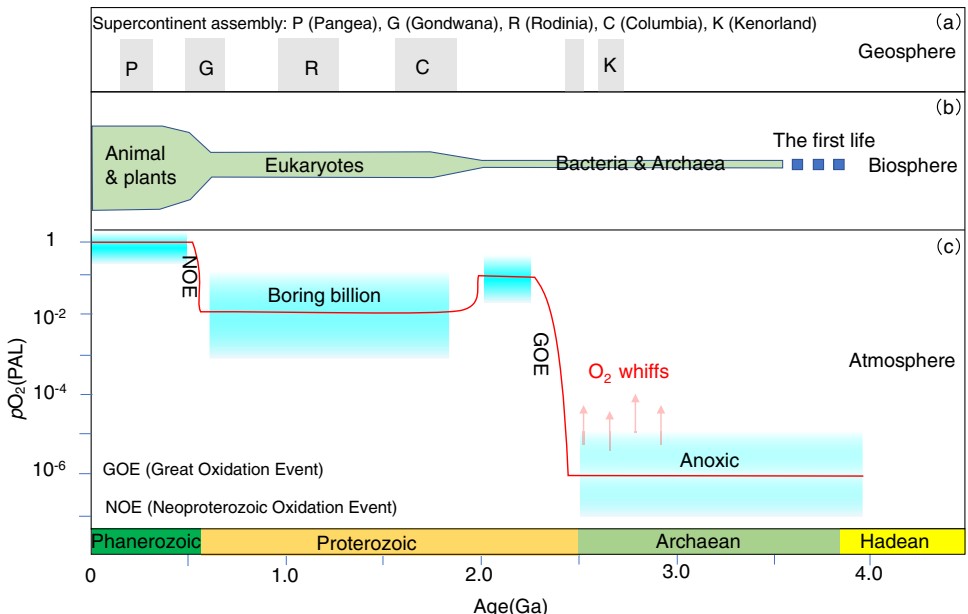

**Fig. 1 | Co-evolution of the Earth system (geosphere, atmosphere, and biosphere) through time. a** Evolution of the geosphere, including the development of the supercontinent cycles[1] (Kenorland, Columbia, Rodinia, Gondwana, and Pangea); **b** Evolution of life within the biosphere[77]; **c** Evolution of atmospheric oxygen levels relative to the present atmospheric level[10]; the blue boxes show a range of atmospheric $O_2$ level inferred from various geological proxy constraints, and the red solid line shows one plausible evolution path of $O_2$ level.

Earth's oxygenation derives from the emergence of oxygenic photosynthesis; however, the GOE likely followed the initiation of this innovation by several hundred million years[16]. The reason for this apparent time lag remains a topic of considerable debate. Various hypotheses rely on the notion that atmosphere oxygen levels are determined by a kinetic balance between $O_2$ production and sinks[17,18]. One possibility is that photosynthetic production of $O_2$ could increase as a result of enhanced nutrients (e.g., phosphorus) availability, and this possibility has been invoked for both GOE and NOE through the weathering of large igneous provinces (LIPs)[19,20]. Yet the rate of organic matter burial ($f_{org}$) has remained approximately constant for most of Earth history[21], rather than showing significant changes that correspond with dramatic, long-term steps in biosphere oxygenation.

Alternatively, it has been proposed that early atmospheric oxygenation (e.g., the GOE) was driven by a secular decline of efficient $O_2$ sinks from the perspective of purely abiotic controls, rather than by an increase in $O_2$ flux[22,23]. Proposed mechanisms that could significantly reduce $O_2$ consumption during GOE include a change in the fraction of subaerial magma degassing[17], mantle oxygen fugacity[24], and/or crustal composition[18,25], all of which would influence the atmosphere oxygen level via the flux and redox state of reductants (gases, minerals, and fluids) emanating from the solid Earth. In general, the redox state of these reductants was controlled by the upper mantle melt[26] and/or its subsequent magma differentiation[27]. It has been suggested that mantle redox did not change significantly through time as evidenced by V/Sc ratios in basalts[28], leading to its frequent dismissal as a key driver of the atmospheric $O_2$ evolution for two decades. Recent studies, however, suggest that the mantle may have gradually oxidized from the Archaean onwards[29,30], resulting in a decline of oxidizable volcanic gases that could have trigged the GOE[23]. Moreover, the rapid formation of felsic continent through plate tectonics could largely reduce the flux of mantle-derived reductants (e.g., $Fe^{2+}$ and $S^{2-}$), ultimately allowing atmospheric $O_2$ to accumulate during the GOE[18]. These many and often competing arguments leave open the need for novel perspectives.

Mafic rocks like basalts act as key probes of Earth's interior and redox state, and their formation and geochemical evolution are directly related to the flux of nutrients and reductants to Earth's surface[31], thus playing a crucial role in both $O_2$ production and sinks. Global data compilations for igneous rocks (e.g., EarthChem) provide much higher temporal resolution of variations in geochemical concentrations throughout Earth history[27,32] compared to geochemical proxies preserved in sedimentary repositories (e.g., Sedimentary Geochemistry and Paleoenvironments Project)[33]. These large datasets for igneous rocks include dozens of elements and ages ranging from 4.0 Ga to the present, as shown in Figs. 2 and 3, with details described in the "Methods". Big data approaches to global igneous geochemistry have been employed previously to explore secular evolution of mantle melts[27], supercontinent cycles[32], and historical trends in plate tectonics[34]. Moreover, integrated analysis of diverse data resources using machine learning, in contrast to separate treatment of each physicochemical property, can lead to more evidence-based predictions about many aspects of Earth system and its relationship with surface environmental change[4,5,35].

With this goal in mind, we focus on reconstructing atmospheric oxygen variation over deep time using machine learning based on global mafic igneous geochemistry big data. Nonetheless, it should be emphasized that rigorous screening of rock samples (e.g., age data filtering) is essential when applying big data approaches to global geodatabases[36]. In this paper, using unsupervised learning methods (including self-organizing map (SOM) and principal component analysis (PCA)), we find that the state of mafic igneous composition evolved chemically through the four major stages, with geochemical transition periods at both ends of the Proterozoic, corresponding with the timing of GOE and NOE, respectively. Second, the supervised methods (e.g., supporting vector regression (SVR)) based on Monte Carlo simulations were applied to the same dataset to "predict" atmospheric oxygen levels since 4.0 Ga quantitatively. Our independent method successfully revealed a high-resolution two-step pattern of increasing atmospheric oxygen level and identified many second-order fluctuations since 4.0 Ga, most of which agree with local observations based on sediment-hosted paleo-oxybarometers. We hypothesize from these observations that secular geochemical evolution of mafic igneous systems, as a result of mantle cooling over time[27],

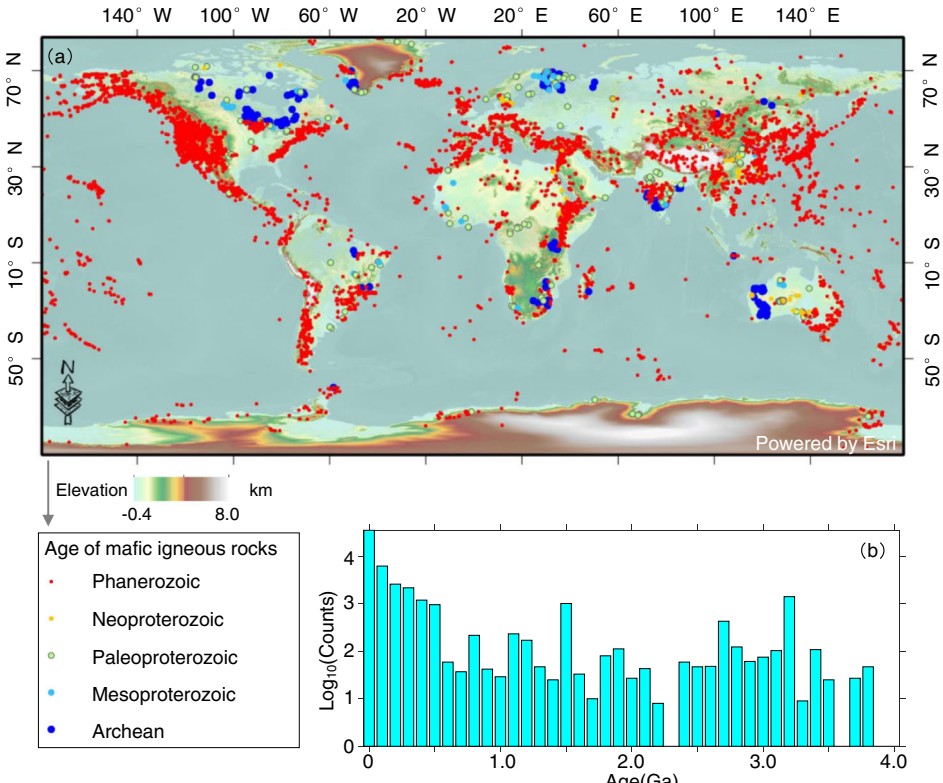

**Fig. 2 | Spatiotemporal distribution of mafic igneous rock samples. a** Spatial distribution of global data compilation of whole rock mafic igneous compositions from the EarthChem, GEOROC, and USGS data repositories. The map was created with ArcGIS 10.8. **b** Histogram of age-frequency distribution of mafic igneous rock samples (age bin = 100 Myr).

could have played an important role in the secular change of atmospheric oxygen levels on Earth.

## Results and discussion
### Secular change of mafic igneous geochemical composition

We employed unsupervised machine learning to examine the first-order pattern of mafic igneous geochemical evolution through time from concentration data for 44 elements. SOM[37] and PCA[38] are the two most widely used unsupervised learning methods. They can effectively reduce a high-dimensional dataset into relatively few representative samples that can be more easily visualized, quantified, and interpreted. In our study, the utility of SOM lies with its ability to delineate major trends, jumps, and clusters in the global igneous geochemical time series dataset. We used PCA to supplement SOM results by investigating the details of temporal variations observed via the first principal component (PCA1). Additional detailed information about the practical implementation of SOM and PCA in this study can be found in the "Methods" and Supplementary information. First, SOM analysis of global mafic igneous data for 44 elements has identified four major classes of mafic composition, reflecting four distinct time intervals since 4.0 Ga (Fig. 4a). Class 1 represents the geological interval from ~4.0 to 2.5 Ga. These data are characterized by high concentrations of highly compatible elements, which are preferentially partitioned into solid phases (e.g., MgO, Cr, and Ni), and low concentrations of incompatible elements, which are preferentially partitioned into the melt (e.g., $K_2O$, $Na_2O$, $P_2O_5$, Th, and U). Class 2 spans ~2.5 Ga to 1.8 Ga and represents a transition period of mafic igneous compositions for most elements during the Paleoproterozoic time. Class 3 spans ~1.8 Ga to 0.7 Ga and represents a relatively stationary state of mafic igneous compositions for most elements during Earth's middle age. Class 4 represents ~0.7 Ga to the present and is characterized by relatively lower concentrations of compatible elements and higher

concentrations of incompatible elements. The secular change in concentration of compatible elements (decrease) and incompatible elements (increase) in mafic igneous rocks reflects degrees of mantle melting controlled by mantle cooling through time[27]. Therefore, the identified four broad steps of mafic igneous geochemistry evolution through SOM analysis reflect the first-order response of decreasing mantle melting degree over the course of Earth history.

A major finding here is the existence of four major stages of mafic igneous geochemistry evolution separated at ~2.5 Ga, ~1.8 Ga, and ~0.7 Ga (Fig. 4a). The first stage, spanning from ~4.0 to 2.5 Ga, is marked by elevated mantle temperatures[39] and thinner crust[40] compared to the present day (Fig. 4c). The second stage, from ~2.5 Ga to 1.8 Ga, overlaps with a major rise of continents, the GOE, and global glaciations, and it coincides with model predictions for early plate tectonics involving hot subduction with shallow slab breakoff[41]. At this time, the continental crust became thicker and more evolved (Fig. 4c). The third stage, from ~1.8 Ga to 0.7 Ga, corresponds with the "boring" billion interval, a period of declining subduction activity and relative lithospheric and environmental stability possibly associated with the incomplete breakup and reassembly of Columbia (also known as Nuna)[42]. The fourth stage, from ~0.7 Ga to present, corresponds to modern cold subduction but with increasingly thinner crust formation and enhanced mantle cooling[43] (Fig. 4c). The geochemical transition period around 0.7 Ga is accompanied or followed by several geologically extreme events that may be interconnected, including the breakup of the Rodinia supercontinent, the NOE, the Snowball Earth events, and the appearance of animal life. This first-order partitioning of mafic igneous geochemical evolution suggests periods of profound change in the Earth system notably near the Archaean-Proterozoic and Proterozoic-Phanerozoic boundaries, with implications for mantle melt, crustal differentiation, atmospheric oxidation, and co-evolving life on Earth.

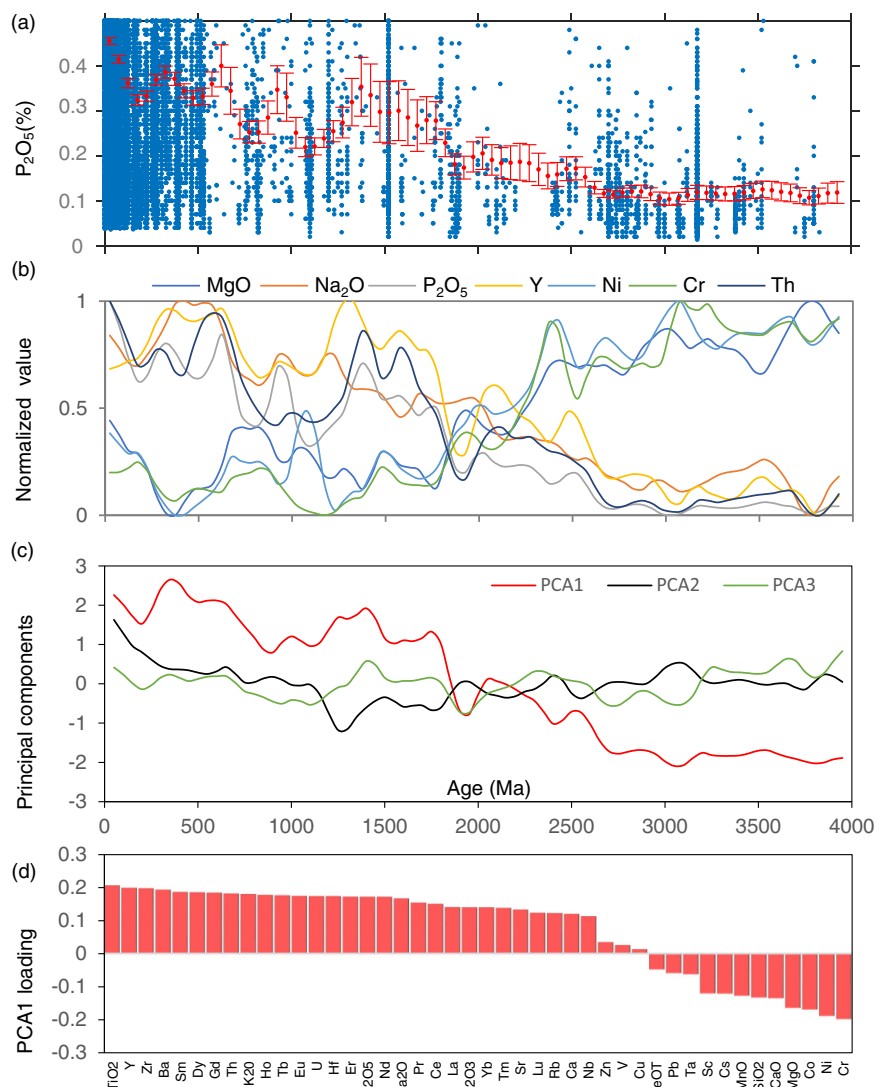

**Fig. 3 | Igneous geochemistry big data analysis. a** Time series record of mean $P_2O_5$ concentration in mafic igneous rocks across the last 4.0 Gyrs; blue circles represent individual values, and red circles are bootstrapped averages with error bars showing 2-SD (standard deviation) uncertainties. **b** Secular compositional evolution of mafic composition through time; time series values of each element's concentration were normalized using max-min scaling algorithm. **c** Principal component analysis (PCA) of mafic igneous geochemistry composition data; the first three principal components (PCA1, PCA2, and PCA3) of mafic igneous geochemistry time series are presented. **d** Factor loadings for each element in the first principal component (PCA1). The sign of a loading indicates whether the variable correlates positively or negatively with the principal component.

## Coupling igneous geochemistry evolution and atmosphere oxidation

Figure 4a shows that the SOM pattern of mafic igneous geochemistry coincides temporally with the classical step rise view of atmospheric oxygen evolution[10,44]. This temporal coincidence of mafic igneous geochemical transitions and atmospheric oxygenations (notably at ~2.5 Ga and ~0.7 Ga) leads us to speculate that deep geochemical processes may have been connected to both of the two great oxidation events on Earth's surface—not just to the GOE[27]. During the earliest stage, high-degree mantle melting resulted in high concentrations of highly compatible elements and low concentrations of incompatible elements in mafic igneous rocks (Fig. 3b), which indicates a high flux of mantle-derived reductants (e.g., $Fe^{2+}$, $S^{2-}$, and Ni) as well as a limited nutrient supply (e.g., $P_2O_5$; Fig. 3a) to Earth's surface. A combination of high reductant concentration and large volumes of mafic rocks dominating Archaean crust may have overwhelmed $O_2$ production fluxes, limiting $O_2$ accumulation until the GOE. Perhaps as a result of gradual mantle cooling accompanied by an abrupt response in degree of mantle melting around 2.5 Ga[27], the rapid decline in concentration of

compatible elements and increase in incompatible elements could reflect a significant decrease of the mantle-derived reductant flux and increase in nutrient supply, respectively, to Earth's surface. This geochemical transition, together with a change in continental crust from mafic to felsic composition[18] and the rise of continents[45], could have shifted the balance between $O_2$ production and sinks toward the production side and therefore collectively triggered the first rise in $O_2$ during the GOE. The second stage of mafic igneous geochemical evolution spans the interval containing the GOE (ranging from ~2.3–2.4 to 2.0 Ga) and persists to ~1.8 Ga. The geochemical transition at ~1.8 Ga, the onset of the third stage, is linked to a possible oxygenation event during the assembly of the Columbia supercontinent[46] and the beginning of the "boring" billion.

Over the subsequent "boring" billion interval, the prolonged stasis of atmospheric oxygen[15] may relate to sluggish evolution of mafic igneous geochemistry and corresponding limitations in nutrient supply and reductant release, thereby maintaining the balance of $O_2$ supply and sinks at relatively low levels for both. For example, sluggish evolution of macronutrients (e.g., $P_2O_5$) during the "boring" billion

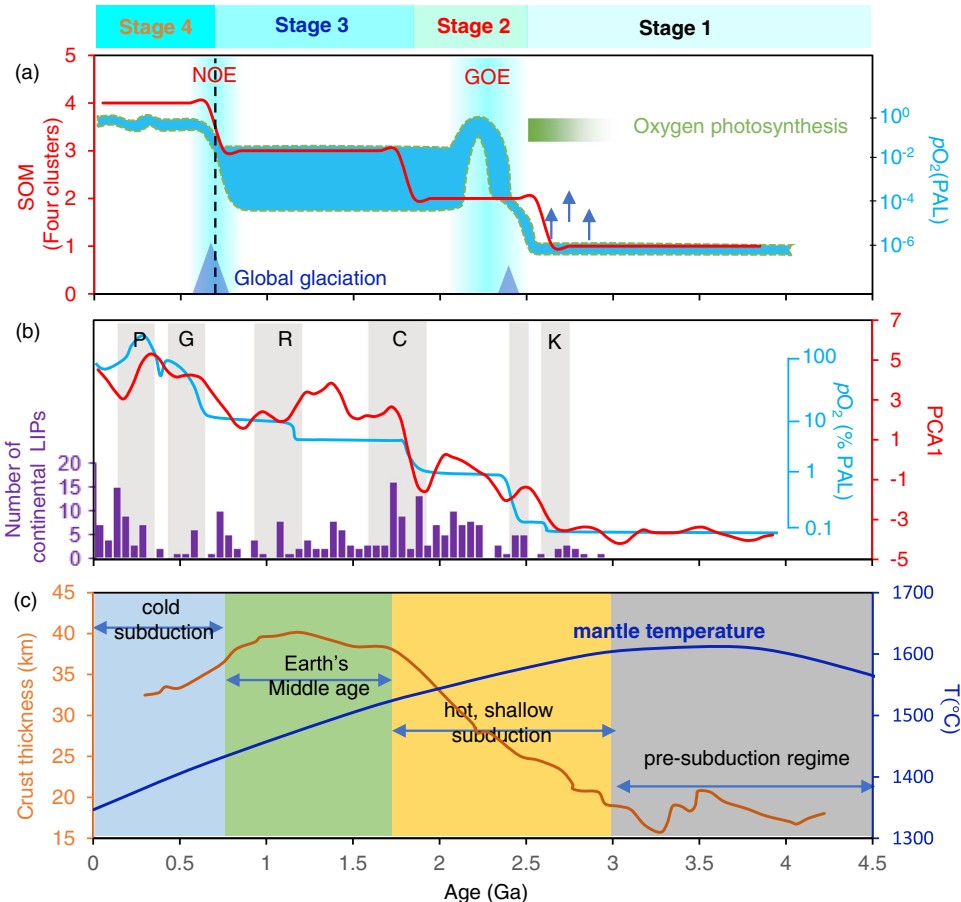

**Fig. 4 | Coupling of mafic igneous geochemical evolution and atmospheric oxygenation. a** Self-organizing map (SOM) classification of mafic igneous compositional evolution through time using four clusters (red line). The blue box shows a "classical two-step" view of atmospheric oxygen evolution[10]. Arrows denote possible "whiffs" of $O_2$ in the Late Archaean. **b** The first principal component (PCA1) of mafic igneous compositional evolution through time (red line), gradual step increases in atmospheric $O_2$ suggested by Campbell and Allen[46] (blue solid line), and numbers of continental large igneous provinces (LIPs)[78]. **c** Four stages of plate tectonics evolution[43]; orange curve represents variation in the thickness of new continental crust through time, and blue line is for secular cooling of mantle temperature.

suggests a limited supply from mantle-derived melts that could have limited primary productivity[47]. Moreover, the temporal coincidence of the igneous geochemistry transition and atmospheric oxygenation patterns around 0.7 Ga has not been reported previously. Tectonically, this geochemical transition corresponds to the onset of a cold subduction style of plate tectonics that could have intensified mantle cooling[43], likely resulting in a second abrupt response to the degree of mantle melt expressed in the -0.7 Ga transition in mafic igneous geochemistry. As such, the increase in continental nutrient inventory (e.g., $P_2O_5$; as shown in Fig. 3a) and a further decline in mantle-derived reductants via secular mantle cooling[31] could have collectively triggered the second significant rise in $O_2$ during the NOE.

Moving forward, PCA of the global mafic igneous geochemical data for 44 elements has identified the most significant principal component, PCA1, which accounts for 78% of the total variance (Supplementary Fig. 6). Overall, the PCA1 pattern, as shown in Fig. 4b, coincides with the SOM pattern but provides more details for mafic igneous geochemical evolution through time. Intriguingly, there is a synchronous second-order relationship between PCA1 geochemical patterns and suggestions of atmospheric oxygen variation (Fig. 4b). First, the short-term return to low oxygen conditions at 2.0–1.8 Ga has been suggested previously by means of Cr isotope data[12,48], although conflicting views on collapse of $O_2$ levels after the GOE remain[49]. In our model, this deoxygenation overlaps with second-order transition in the PCA1 pattern of evolving mafic igneous geochemistry (Fig. 4b)—specifically correlating with peaks in

concentration of highly compatible elements (e.g., MgO, Ni, and Cr) and troughs of incompatible elements (e.g., $P_2O_5$). It has been suggested that the second-order variations in mafic igneous geochemical concentrations also reflect changes in mantle potential temperature and melting degree[32], with termination of the GOE linked to a 2.06 Ga LIP event[50], as shown in Fig. 4b. Therefore, we speculate that the episodically amplified mantle activity could have increased the flux of mantle-derived magmas and thus the flux of oxygen-consuming compounds (e.g., $Fe^{2+}$, $S^{2-}$, and reduced volcanic gases), perhaps contributing to decreases in surface oxygen levels over relatively short time intervals.

Subsequently, the increased delivery of limiting nutrients through the enhanced weathering of LIPs in combination with reduced frequency of volcanism during assembly of the Columbia supercontinent could have favored a recovery of oxygen level[46], which also coincides with a remarkable increase in concentrations of incompatible elements (e.g., $P_2O_5$) as well as in the SOM and PCA1 patterns of mafic composition at -1.8 Ga (Fig. 4b). In addition, the PCA1 pattern of mafic composition suggests an obvious second-order rise at -1.4 Ga, overlapping with the possible transient oxygenation event suggested by multiple sediment-host geochemical proxies (e.g., chromium isotopes)[44,51,52]. While both SOM and PCA1 patterns of mafic composition discussed here are semi-quantitative, and uncertainties remain, we infer that the second-order fluctuations in atmospheric $O_2$ level can be tied to second-order variations in mafic igneous geochemistry evolution.

## Reconstructing atmospheric O₂ levels using machine learning

Although there is a strong correlation between the mafic igneous geochemical concentrations and atmospheric oxygen level through time, the specific cause-and-effect relationship among them and the underlying biogeochemical model remain unknown. In principle, variation in atmospheric oxygen level through time is determined by the balance between the photosynthetic supply of oxygen and its consumption through oxidative reactions. Thus, the rate of $O_2$ change in the atmosphere can be formulated as[18,53]:

$$\frac{dm_a}{dt} = J_{in}^{O_2} - J_{out}^{O_2}, \tag{1}$$

where $t$ represents time, $m_a$ represents mass of $O_2$ in the atmosphere, $J_{in}^{O_2}$ represents the net production rate of $O_2$ (through aerobic photosynthesis and respiration), and $J_{out}^{O_2}$ is the consumption rate of $O_2$ through reductants derived from Earth's interior. At times of low oxygen levels, $J_{out}^{O_2}$ scales linearly with $m_a$–specifically, $k_{out}m_a$, where $k_{out}$ represents the efficiency of oxidation closely related to total rate of magmatic output of direct and indirect oxygen-consuming agents. For example, the latter includes Ni supply, which can control levels of methanogenesis[54], an oxygen sink. The production of $O_2$ ($J_{in}^{O_2}$) is tied to continental nutrient inventory[47], organic carbon burial rate, and global carbon inputs to the ocean-atmosphere system[18]: $J_{in}^{O_2} = \lambda_p J_{out}^{CO_2}$, where $\lambda_p$ encapsulates the controls that regulate primary productivity through nutrient supply from continents (most notably, mafic igneous rocks)[31], and $J_{out}^{CO_2}$ is the sum of the metamorphic and mantle inputs[18]. Therefore, Eq. (1) can be further written as:

$$\frac{dm_a}{dt} = \lambda_p J_{out}^{CO_2} - k_{out}m_a. \tag{2}$$

As both $k_{out}$ and $\lambda_p$ are multivariable functions whose inputs consist of multiple members (e.g., various oxygen-consuming and bio-essential compounds, respectively) changing with time, it is unrealistic to obtain an analytic mathematical expression of $m_a$ through integration of the differential Eq. (2). Conceptually, however, atmospheric oxygen levels should track mafic compositions because the latter modulate the balance of early $O_2$ sources and sinks[31].

Traditional estimates of ocean redox and atmospheric $O_2$ content by association commonly involve an inversion process that uses the concentration or isotopic composition of redox sensitive trace elements (e.g., Cr, Mo, U, Re, and V) in marine sedimentary rocks to guide paleo-oxybarometers[10,48]. Nevertheless, preservation bias is inescapable in sedimentary rocks, and the resulting temporal coverage of suitable sedimentary archives is patchy, especially in the Precambrian, leading to an interpolated $O_2$ curve across the last 4.0 Gyrs with large uncertainties. As such, robust quantitative estimates of atmospheric $O_2$ levels require data compilation of paleo-oxybarometers on a global scale (notably, higher temporal resolution) and better evidence-based predictions using more integrated approaches[55].

Here, we instead employ a forward process to reconstruct atmospheric oxygen level over deep time using mafic igneous geochemical composition data. Building on the compilation of geochemistry big data from global mafic igneous rocks, we can explore oxygen levels using the advanced integrated approach–that is, data-driven supervised machine learning[5,6] without any specific physicochemical model. For simplicity, we assume a nonparametric multivariate transform, directly relating $m_a$ to all mafic igneous geochemical compositions, which can be written in the general form:

$$m_a = F(\mathbf{X}) + \epsilon, \tag{3}$$

where $F$ represents some fixed but unknown function of $\mathbf{X} = (x_1, x_1, \ldots, x_n)$ (i.e., element contents), and $\epsilon$ represents a random error term independent of $\mathbf{X}$. Supervised machine learning methods (e.g., neural network, random forests, and support vector machine) are often used for nonparametric nonlinear regression based on labeled training sets. The training dataset includes inputs and correct outputs, allowing the computer to learn a function ($F$) that maps input predictor ($\mathbf{X}$) to the output prediction ($m_a$) via optimization algorithms. As such, supervised machine learning methods are able to fit a large number of complex functions without strong assumptions about their specific forms.

We first use a SVR[56] to reconstruct atmospheric oxygen variation over deep time based on global mean mafic igneous geochemistry composition for 44 elements. Specifically, SVR learns a function ($F$) from the training label pair (i.e., known contents of elements and $O_2$ within limited time intervals), and once training is done it can be used further to map the known element contents ($\mathbf{X}$) onto the output ($m_a$), namely, unknown $O_2$ contents for the remaining time intervals. Thus, preparation of training samples is the most important step in prediction of atmospheric $O_2$ content when using supervised machine learning. In our case, the consensus on the general two-step rise model for atmospheric oxygenation (Fig. 1) can be used as prior knowledge for building labels, with large uncertainty in the estimated magnitudes of change–from $10^{-9}$ to $10^{-5}$ PAL pre-GOE and $10^{-4}$ to $10^{-1}$ PAL during the "boring" billion interval from ~2.0 to 1.0 Ga and $10^{-1}$ to ~1 PAL from ~0.5 Ga to present. In order to propagate this uncertainty, a Monte Carlo simulation (MCs) algorithm was used to prepare the training dataset and make predictions. Specifically, the scheme for preparing the training label pairs involves labeling random numbers of ages (bins) with stochastic $O_2$ contents within the confidence intervals of three recognized $O_2$ base levels, while using corresponding geochemical concentrations of all 44 elements as predictors. More details about the practical training, validation, and implementation of machine learning in this study are described in the "Methods".

As shown in Fig. 5b, the trained SVR model, with mean average accuracy ($R^2 > 0.9$) and mean square error ($<0.01$) (Supplementary Fig. 7) in the test data validation after 1000 MCs, recognized the first-order trend–i.e., a two-step rise of atmospheric oxygen level spanning Earth history over roughly the last four billion years. The two transition periods of sharp $O_2$ increase during time intervals of 2.5–2.1 Ga and 0.8–0.6 Ga coincide with the previously recognized timing of GOE and NOE[10], respectively. Even if we prepared the training data using large uncertainties for labeling $O_2$ levels post-0.5 Ga and pre-2.5 Ga through MCs (without any $O_2$ labels during 2.5–0.5 Ga), this two-step rise temporal pattern of atmospheric $O_2$ evolution path can be recovered through our machine learning modeling (Supplementary Fig. 8). This model estimates a relative ~1000-fold increase of atmospheric $O_2$ level during GOE and ~100-fold during NOE (Fig. 5b), both of which are consistent with previous estimates from sediment-hosted paleo-oxybarometers[10,17]. We conclude from this agreement that evolving mantle melts could have contributed to the main events of atmospheric oxygenation by modulating the balance of early $O_2$ sources and sinks.

## A high-resolution pattern of atmospheric O₂ variation

Our model provides a higher-resolution temporal pattern of atmospheric oxygen variation, revealing possible second-order fluctuations along with the first-order trends of the classical model. More importantly, as shown in Fig. 6, most of these second-order fluctuations coincide with possible transient oxygenation and deoxygenation events supported by multiple lines of evidence preserved in sediment-hosted paleo-oxybarometers[44,51,55]. First, our results (Fig. 6) indicate Archaean oxidation events (AOEs) at ~3.0 Ga and ~2.5 Ga. The ~3.0 Ga event coincides generally with the emergence of oxygenic photosynthesis[16], and the ~2.5 Ga event is linked to the whiffs of oxygen as discussed previously[57]. Multiple lines of evidence from sedimentary records are consistent with such events (reviewed in ref. 58) and

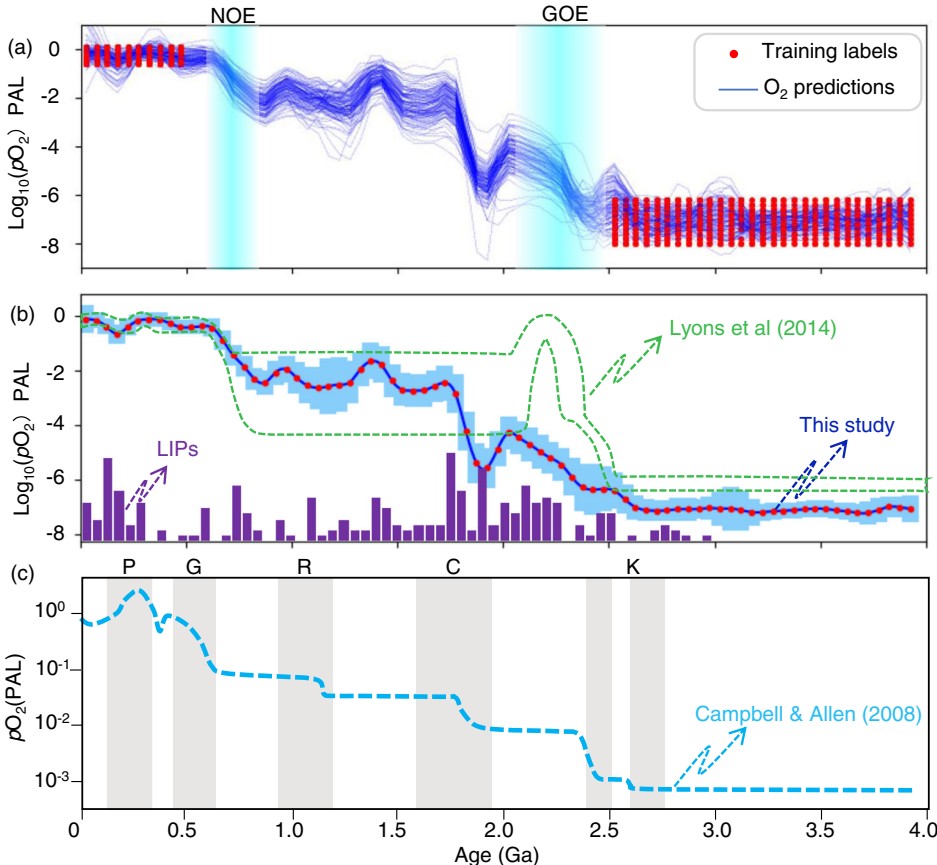

**Fig. 5 | Quantitative prediction of atmospheric O₂ content using machine learning. a** All curves of support vector regression (SVR) modeling of atmospheric O₂ content using global mafic igneous geochemistry data through Monte Carlo simulations ($n = 100$), with random training label pairs selected pre-2.5 Ga and post-0.5 Ga. **b** Estimated atmospheric O₂ variation through time using mean value of 1000 simulations; the error bar shows 2 standard deviation (2-SD) uncertainties, and the green box represents the atmospheric O₂ model of Lyons et al.[10]. **c** Multi-step rise model of atmospheric O₂ level proposed by Campbell and Allen[46], which is linked to supercontinent formations.

suggest that the GOE was the culmination of an extended period of transient oxidation[10,59].

The background atmospheric O₂ level prior to the 2.5 Ga event was less than $10^{-7}$ to $10^{-5}$ PAL based on the conventional interpretation of sulfur mass-independent isotope fractionations (S-MIF) found in the sedimentary record at that time (Supplementary Fig. 14). Except for a short-term return to low pO₂ after an AOE at ~2.5 Ga, however, our model predicts that atmospheric O₂ levels exceeded $10^{-5}$ PAL at ~2.3 Ga and suggest a continuous climb in pO₂ to $10^{-4}$ PAL by ~2.1 Ga. While our result agrees with previous estimates of the onset of the GOE span 2.5–2.2 Ga[13,60], it does not seem to support the O₂ oscillations suggested by low S-MIF signals occurring from ca. 2.45 to 2.2 Ga[14]. However, a higher temporal resolution (<50 Myr bin) of global igneous geochemical composition is required to explore the possibility of these oscillations through the GOE with confidence when using our machine learning model.

During the "boring" billion, the prediction of atmospheric oxygen levels varies within a range of $10^{-3}$ to $10^{-1}$ PAL, and this range is in agreement with previous estimates[15]. As mentioned above, whether O₂ levels were persistently low for a billion years following the GOE remains controversial[15,51]. Our results suggest that the atmosphere during the "boring" billion maintained intermediate but still low O₂ levels. However, this prolonged period of low O₂ could have been punctuated by episodes of elevated O₂ relative to a lower baseline, perhaps analogous to the whiff events leading up to the GOE. Such episodic features are suggested at ~1.8 Ga, ~1.4 Ga, and ~1.1 Ga in Fig. 6 but with atmospheric O₂ estimates <0.1 PAL. At face value, both peaks can be explained by increases in concentration of macronutrient

phosphorus and decrease of reductants concentration in mafic rocks, and ~1.8 Ga and ~1.1 Ga events are coupled with supercontinent assemblies.

The possibility of increased atmospheric O₂ level around 1.8 Ga is consistent with the disappearance of iron formations at that time (Fig. 6c)[61], as their formation was favored high dissolved $Fe^{2+}$ in the oceans under widely anoxic conditions. Moreover, the ~1.8 Ga oxygenation event coincides temporally with a volume increase in maximum organism size[62] and roughly with the first fossil eukaryotes (Fig. 6). The transient oxygenation event at ~1.4 Ga is suggested by several lines of evidence from sedimentary geochemical proxies (e.g., $\delta^{53}$Cr, $\delta^{98}$Mo, and U)[52], although debate about the specifics remains[44,63]. Additional evidence consistent with possible increasing atmospheric O₂ around 1.1 Ga includes suggestions of increased sulfate concentrations in the oceans[64] and a pronounced peak in Cr isotopic composition of carbonates[65] and Re concentrations in organic-rich shales[51,66]. Although this "event" is still only suggested by the sediment-hosted geochemical data[44,51], its agreement with our findings should spawn additional work.

The supervised learning quantitatively estimated local lows of oxygen content at ~1.9 Ga, ~0.9 Ga, ~0.6 Ga, and ~0.2 Ga, with the oldest previously observed in the PCA1 pattern (Fig. 4b). Although there is no "smoking gun" evidence for deoxygenation at ~0.9 Ga, it has been reported recently that the early Neoproterozoic ocean maintained low oxygen levels as inferred from Se/Co ratios measured in sedimentary pyrites[67] and phosphorus-limited conditions[68]. We posit that these possible deoxygenations, particularly those following the GOE, are closely related to decrease in flux of nutrients supply (e.g., P₂O₅) and

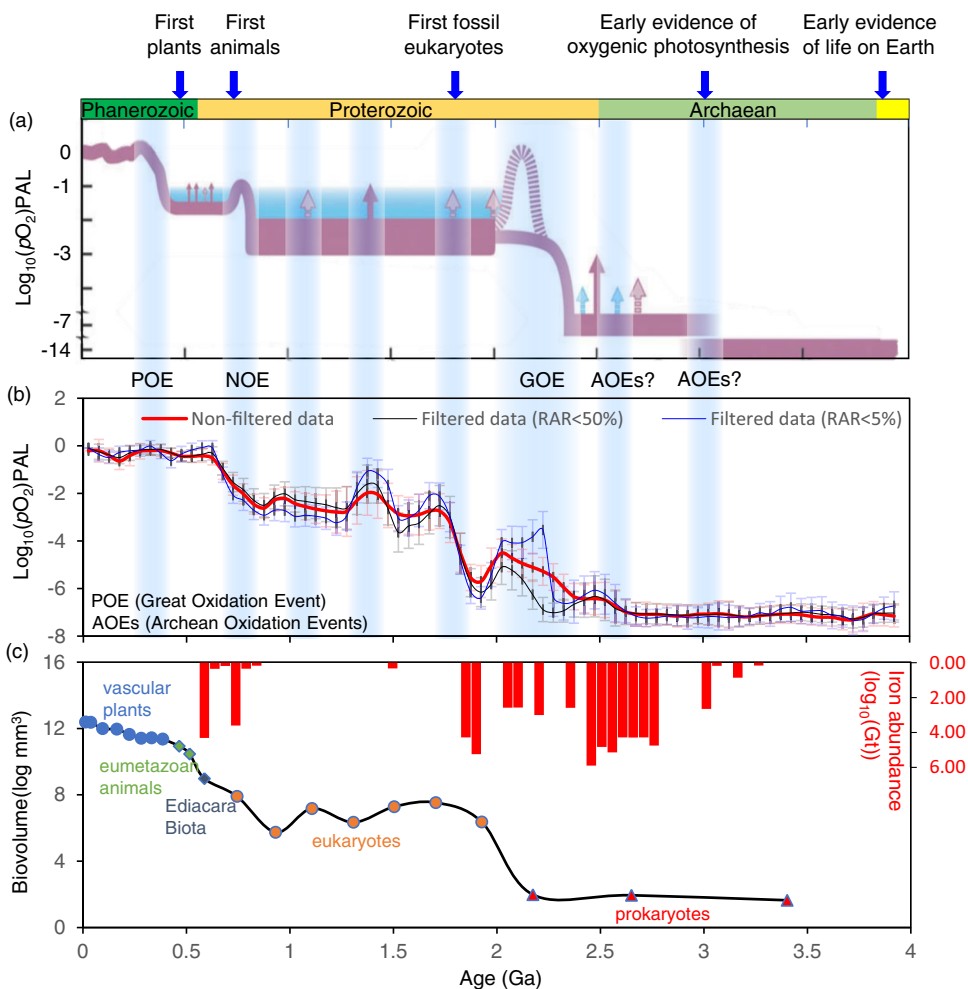

**Fig. 6 | Comparison of the emerging O₂ model to predictions based on sedimentary geochemical proxies. a** Atmospheric O₂ levels across the last 4.0 Gyrs manually reconstructed by multiple lines of evidences preserved in sediment-hosted paleo-oxybarometers (more details can be found in ref. 44). Solid red arrows denote possible transient increases in O₂ for which geochemical evidence exists; pink dashed arrows indicate less certain events. **b** Atmospheric O₂ levels across the last 4.0 Gyrs computationally reconstructed by machine learning using global mafic igneous geochemical data after been filtered using the relative age range[36] (RAR < 5% and 50%); error bars show 2-SD (standard deviation) uncertainties. **c** Temporal variations in the maximum biovolume over the Earth's history[62], and iron formation abundance[79].

increase of oxygen-consuming agents related to enhanced mantle activity (e.g., LIPs). Moreover, our results suggest a short-term return of relatively low O₂ levels after the NOE (see Fig. 6) likely linked to a significant decrease in P₂O₅ concentration in mafic rocks (Fig. 3a). This deoxygenation event at ~500 Ma may be consistent with previous arguments asserting that O₂ remained relatively low or returned to low values during at least the first part of the Paleozoic[44,69]. Following this deoxygenation event, our results suggest a climb to the highest levels of atmospheric O₂ at 0.5–0.3 Ga, known as the Paleozoic Oxidation Event (POE)[69], which can also be attributed to an increase of P₂O₅ concentration in mafic rocks (Fig. 3a).

In addition, the possible transient oxygenation events suggested at ~2.5 Ga, ~1.8 Ga, ~1.1 Ga, and ~0.4 Ga are temporally coupled with supercontinent assemblies (e.g., Kenorland, Columbia, Rodinia, and Pangea; see Fig. 5b). This observation is consistent with the hypothesis of Campbell and Allen[46] that supercontinent assembly contributed to the rise of atmospheric O₂ through the enhanced weathering of continental phosphorus inventory. Interestingly, the two most remarkable increases of atmospheric O₂ level (i.e., GOE and NOE) did not correspond to supercontinent assembly but do coincide well with igneous geochemical transitions as shown in Fig. 5, suggesting that geochemical evolution of mantle-derived magmas could have played a more critical role in modulating the balance of early O₂ sources and sinks.

As an additional step, we have tested the sensitivity of our oxygen reconstruction to different ML algorithms. As shown in Supplementary Fig. 12, the artificial neural network (ANN) and random forests (RF) methods recovered an evolutionary pattern of O₂ similar to SVR. RF modeling provided feature importance of all 44 elements during ML modeling, as shown in Supplementary Fig. 13, illustrating how each element influences the predictions of atmospheric oxygen level. We note that most of (both highly negatively and positively) correlated elements identified in Supplementary Fig. 4 make equally significant contributions to the two-step rise pattern of atmospheric O₂. The evolution patterns for K₂O, Na₂O, and MgO inherently reflect the transition of mantle melting from high to low degree. This finding also suggests that both nutrients (e.g., P and Si) and trace elements (e.g., Co, Ni, and Cr) from mantle-derived magmas could have played important roles in atmospheric oxygenation owing to their effects on the biosphere[70].

Our model may underestimate O₂ production, as this production in our model is linked only to the flux of nutrient supply from mafic igneous rocks and not recycled from sedimentary sources. For example, as seen in Fig. 6, the ML predictions in this study did not completely recover the "oxygen overshoot" during the GOE even though we added training labels of O₂ level during this period (Supplementary Fig. 9). There must therefore exist other drivers that could amplify the

predictions of our model, including but not limited to increased volcanic $CO_2$ input (related to tectonic transition)[18] and/or deep sequestration of organic carbon[71]. In this regard, an amplification of our $O_2$ predictions can be expected (at least) during GOE as linked to rapid rise of continents and major glaciations that increased nutrient inputs and net sequestration of organic C, but we can also predict increased volcanic $CO_2$ emissions due to sediment-enhanced subduction activity[72].

Our model also underestimates $O_2$ sinks, as it does not include processes like organic matter remineralization in the water column and weathering of ancient fossil organics in rocks and aerobic respiration, especially during the Phanerozoic. Moreover, mantle cooling increased the oxidation state of both mantle melts and equilibrated volatile species[73], and the latter could amplify consumption of $O_2$. Again, our overarching hypothesis is that mafic geochemical transitions as a result of secular mantle cooling could play an important role in the rise of atmospheric oxygen. Overall, given that mafic igneous geochemical data alone can quantitatively recover the classic two-step rise model of atmospheric $O_2$ level, as well as the order-of-magnitude of $O_2$ increase previously predicted for the GOE, NOE, and even the POE, we can conclude that atmospheric oxygenation may therefore be at least partly a natural consequence of mantle cooling and specifically played the role of evolving mantle melts in modulating the balance of early $O_2$ sources and sinks.

### Remarks on data-driven scientific discovery
Data-driven machine-learning algorithms with big data provide an additional technique to explore Earth's long-term evolution. An essential benefit of data-driven discovery models is their potential to reveal correlations among massive dataset and whether or not there are causation and thus mechanistic drivers nested within these relationships. Because these datasets can be diverse, high-dimensional, and noisy, machine learning algorithms are needed to handle big datasets to expose previously hidden patterns. Given that applications of artificial intelligence in Earth science have not yet reached the level of cognitive intelligence (i.e., understanding mechanism) and often suffer from a black box problem of ML algorithms (lack of transparency), we emphasize the need to blend human learning and artificial intelligence when searching for the causation that underlies correlation.

The main benefits of data-driven discovery modeling, as demonstrated in our study, also include transformation from known questions seeking unknown answers to seeking unknown questions and unknown answers[2,3]. For example, the two-step rise model of atmospheric oxygen is well known, but how Earth's atmosphere became oxygenated remains a topic of considerable debate. In the present study, the unsupervised machine learning of global igneous geochemical time series data recognized three major stages of its secular evolution, which correlate well with two-step increases in atmospheric oxygen levels over time. This synchronization helps us link the two-step rise and perhaps second-order fluctuations in atmospheric $O_2$ level to deep Earth geochemical processes through mantle cooling and magmatism systems (see also ref. 27).

At the same time, the observed relationships raise previously unknown questions about correlation versus causation and specifically whether global igneous geochemical time series data can indeed quantitatively predict atmospheric $O_2$ levels over deep time. By using supervised machine learning, we have produced a higher-resolution temporal pattern of variation in atmospheric oxygen that has not only successfully and independently reconstructed the classic two-step rise model but also suggested more detailed fluctuations. Specifically, our approach offers mechanistic windows to possible drivers of dynamic Precambrian oxygen levels, while at the same time revealing possible second-order oxygen variability. This second-order fabric is comparable to previously suggested but still poorly constrained

proxy-based records of shorter-term $O_2$ increases and decreases in the oceans and atmosphere. These findings amplify the likelihood that planetary interior processes are critical in atmospheric oxygenation and highlight the need for a better understanding of the roles of planetary interiors as we seek other "Earth-like" planets. Despite the many remaining unknowns, however, machine learning techniques and recent advances in big data and artificial intelligence offer exciting new opportunities for exploring Earth's early evolution and its relevance to other planetary systems.

## Methods
### Data material
Large-scale community geo-data sources, such as EarthChem, GEOROC, PetDB, and Macrostrat are especially relevant for promoting data-driven discoveries in statistical petrology, geochemistry, mineralogy, and sedimentology. Here, we have compiled global mafic igneous composition data from the EarthChem data repository (http://portal.earthchem.org/, assessed Feb. 2022), which includes PetDB, GEOROC, NAVDAT, and USGS databases. This new database contains ~54,000 whole-rock analyses of major, trace, and rare earth elements from mafic igneous rocks (mainly basalt, gabbro, diabase, and tholeiite) with chemical compositions of 43–51% $SiO_2$ and MgO < 18%. In addition, we have eliminated outliers in the geochemical concentration data for each element by using a mean ± $3\delta$ threshold during the Archaean, Proterozoic, and Phanerozoic (Supplementary Fig. 2). Given that most trace element data tend to follow lognormal distributions (Supplementary Fig. 1), a logarithm transformation was applied to the data for each element before outlier filtering. The spatiotemporal distribution, as shown in Fig. 2, suggests that such mafic rock samples cover most of the continents throughout the history of the Earth. Specifically, the compilation data includes mafic rock samples within most age bins (~95% and ~92% for 100-Mys and 50-Myr bins, respectively) ranging from 3.8 Ga to the present. Therefore, the updated database is sufficiently large and tectonically diverse to capture a global signal of mafic igneous geochemistry composition. However, the mafic rock samples have a heterogenous age distribution, with several peaks (Fig. 2b) linked to either crustal preservation bias (oversampling of some periods) or biases tied to age availability (e.g., data points at 1521 Ma and 3175 Ma with large age range/uncertainties).

In order to minimize the issues of spatial or temporal oversampling and data points with large age uncertainties, we used the weighted bootstrap sampling method of Keller and Schoene[27] (https://github.com/brenhinkeller/StatisticalGeochemistry) to calculate the time series of global mean mafic igneous geochemistry composition. The emerging curves of geochemical concentration time series (Fig. 3b) are similar to that of Keller and Schoene but show more details and fewer uncertainties because of the expanded database. We then used the relative age range (RAR) method[36] (RAR = (MAX AGE − MIN AGE)/AGE * 100) to systematically investigate the influence of age uncertainties on igneous geochemical time series patterns when using the weighted bootstrap sampling method. Only 31% and 57% of the samples recorded in the EarthChem Portal have a reported age with a RAR of less than 5% and 50%, respectively. We adopted a more rigorous sample screening scheme (data filtered with RAR < 5% and <50%) to generate time series records of mean mafic igneous geochemical concentration. Supplementary Fig. 3 shows that the patterns for RAR filtered and non-RAR filtered data have consistent time-varying trends (including peaks and troughs) and do not exhibit notably different peak positions (although there are differences in amplitude). This result confirms that the weighted bootstrap sampling method has alleviated the effects of data points with large age uncertainties in time series data preparation.

Cluster analysis was used to explore the correlation matrix for mafic compositions spanning 44 elements. As shown in Supplementary Fig. 4, interesting and distinct clustering emerges: strong

correlations are observed in concentrations of highly compatible elements (including Co, Cr, Ni, and MgO) and incompatible elements (e.g., $P_2O_5$, $K_2O$, $Na_2O$, U, Th, etc.), while these compatible and incompatible elements show remarkable negative correlation. According to the efficiency of mantle convection, long-term cooling of mantle results in decreasing degrees of the mantle melt fraction and more felsic crust through time[27]. Therefore, mafic rock samples record a fluctuating decrease in compatible element concentration (e.g., MgO, Cr, Ni, and Co) and a corresponding increase in incompatible element concentration values (e.g., $K_2O$, $Na_2O$, $P_2O_5$, Th, and U) through time (Fig. 3b). As many bio-essential elements are effectively incompatible, and reductant elements are compatible in the mantle, the secular change in composition of mantle-derived melts through Earth history, as a consequence of secular mantle cooling, could provide a fundamental control on atmospheric oxygenation through nutrient supply and reductant outputs[31]. Further, it has been suggested that the second-order temporal variations of mafic igneous mean geochemical concentration also reflect changes in mantle potential temperature and degree of mantle melting, with profound implications for mantle dynamics and/or supercontinent formation[32].

### Unsupervised machine learning

We used unsupervised machine learning (including SOM and PCA) to reduce the high dimensionality of global geochemistry big data (spanning 44 elements) and to explore the first-order relationships of mafic igneous geochemistry evolution through time. SOM is an unsupervised machine learning algorithm that draws on the basis of neural networks using competitive learning rather than the error-correction learning[37]. It maps multidimensional data points into a lower dimensional space, where neurons are topologically ordered and similar input data are projected onto nearby neurons (i.e., by prototype vector) on the map, thereby preserving the topological structure of the data. SOM analysis mainly involves two stages, training and mapping phases. In this study, the goal of SOM analysis is to represent igneous geochemistry big data with 44 dimensions as one-dimensional time series data. Several parameters, such as iteration step, learning rate, and structural function should be specified by the user prior to training the network. Nevertheless, they have little influence on patterns produced on the SOM maps.

We selected time series data of all 44 elements as input data, and a min-max algorithm was used for data normalization. The unified distance matrix (U-Matrix) (Supplementary Fig. 5a) and component plots (Supplementary Fig. 5b) were obtained for visualization of SOM groups to obtain a quantitative clustering. As a result, we can observe several distinct clusters of "winning" neurons in component plots distinguished by borders in the U-Matrix scalogram. However, manual selection of the clusters could be arbitrary and lead to artificial results. Therefore, K-means algorithm is often used to cluster the prototype vectors instead of the original SOM data, and here the optimal number of SOM clusters is determined ($N = 4$) by using Davies-Bouldin (DB) analysis[74]. As a result, SOM analysis identifies four broad steps of mafic igneous geochemistry evolution through time (Fig. 4a), with transitions at ~2.5 Ga, ~1.8 Ga, and ~0.7 Ga. Nevertheless, more clusters can be obtained when we need to explore the second-order partitioning of mafic igneous geochemistry evolution through time. As shown in the component plots in Supplementary Fig. 5b, compatible elements (e.g., MgO, Cr, Ni, and Co) show strong contribution to class 1 and make weak contribution to class 4 (Supplementary Fig. 5c), while incompatible elements (e.g., $K_2O$, $Na_2O$, $P_2O_5$, Th, and U) do the opposite. We performed the SOM analysis of mafic igneous geochemical data using the compiled MATLAB code based on the open SOM Toolbox available at http://www.cis.hut.fi/projects/somtoolbox/.

PCA is one of the most commonly used methods for dimensionality reduction enabling us to project high-dimensional data onto the first few principal components, while preserving important data information. For example, the first principal component can be defined as the direction that maximizes the variance of the projected data. The principal components can be explained in terms of associations among variables (e.g., by using eigenvectors or loadings) that are not apparent without statistical analysis. In this study, the goal for PCA was to extract the important temporal variations within a large igneous geochemical time series dataset (spanning 44 elements) and to express this temporal information using the first few principal components.

As shown in Supplementary Fig. 6c, PCA1 accounts for 78.2% of the total variance, while PCA2 accounts for only 7.0%, with the others accounting collectively for 14.8%, suggesting that most information in the mafic geochemical data can be represented by PCA1. As shown in Fig. 3c, the PCA1 pattern shows an overall increasing trend through time, which reflects the secular changes in mantle potential temperature and degree of mantle melting, while the PCA2 and PCA3 patterns show less agreement with any geological time series trend. More importantly, the temporal pattern of PCA1 is in agreement with the classical two-step view of atmospheric oxygen evolution. The magnitude of loading (Supplementary Fig. 6d) represents the covariances/correlations between the original variables (elements) and the unit-scaled components, indicating the contribution of each element to the principal component. Specifically, as shown in Fig. 3d, incompatible elements (e.g., $K_2O$, $Na_2O$, $P_2O_5$, Th, and U) have positive loading values for PCA1, whereas compatible elements (e.g., MgO, Cr, Ni, and Co) have negative loading values. Most of the selected elements make equally significant contribution to the PCA1 pattern, with $TiO_2$, Cr, Ni, and Y being slightly more important.

### Supervised machine learning

Framing the estimate of $O_2$ content as a supervised machine learning regression problem, we can label $O_2$ levels in specific time intervals using the published $O_2$ model with uncertainties and employ multiple lines of evidence for oxidation as input. The goal of using machine learning is to parameterize the complex relationship between such training labels and igneous geochemical proxy data (in the training data process) to quantitatively estimate $O_2$ content over the course of Earth history. SVR is a regression algorithm that draws on support vector machine (SVM)[56] with the basic idea of separating between classes in a dataset within an optimal hyperplane that maximizes boundaries between classes. SVM and SVR have been widely used for classification and regression problems, respectively. Data used to train and evaluate supervised machine learning algorithms are generally divided into training set and testing set. In this study, all data were scaled between 0 and 1 prior to inputting into the SVR, and a Gaussian radial basis function kernel was used because this is a reasonable first choice for most applications. The performance of SVR is also controlled by the value of the penalty term ($C$) and parameter of the kernel function—for example, width ($\gamma$) of the radial function as used in this study. Note that very large penalty ($C$) or width ($\gamma$) may lead to over-fitting and poor generalization capability, even if it shows high accuracy in data training.

A five-fold cross-validation approach was used in this study to avoid the over-fitting problem in data training and to determine the optimal $C$ and $\gamma$ values. In the stochastic modeling, we undersampled the training samples from the originally prepared (input and output) dataset (within the periods of 0–500 Ma and 2500–4000 Ma) for each SVR training and then divided them into five subsets of the same size. In each training round, one subset was selected successively as the test dataset and the rest as training datasets. The above process was repeated five times so that all subset data could be included for testing and validation. The final performance was evaluated by averaging results from all repeated training rounds. We compiled Python codes to implement the reconstruction of historic atmospheric $O_2$ levels

using supervised machine learning based on Sklearn library (https://scikit-learn.org/stable/install.html).

Supplementary Fig. 7a shows an example of automatic determination of the optimal $C$ and $\gamma$ values in SVR modeling for one MC. The root-mean-square-error (RMSE) generally converge to 0.01 in stochastic SVR modeling, as shown in Supplementary Fig. 7b. Here, the effects of uncertainties in $O_2$ prediction when using $O_2$ different base level as training labels have been evaluated. The results in Supplementary Fig. 8 suggest that using the $O_2$ base level with different error ranges post 0.5 Ga and pre 2.5 Ga did not affect the final prediction of temporal variation of atmospheric $O_2$ level, nor the order-of-magnitude of $O_2$ increase during the GOE and NOE.

We have conducted numerous tests for SVR predictions using different datasets, including filtered datasets with RAR < 5% and < 50%, the igneous geochemical time series datasets with 50-Myr and 100-Myr resolutions, using published and our updated mafic rock databases, with the raw and the centered log-ratio (CLR) transformed datasets. First, as shown in Fig. 6b, the predicted patterns for atmosphere $O_2$ level between RAR filtered and non-RAR filtered datasets did not show notable differences in temporal trends, except for the slight differences in amplitudes, most of which fall in the uncertainty range provided by MCs. As shown in Supplementary Fig. 10a, the atmospheric $O_2$ modeling is robust with or without the less relevant elements (i.e., those with low correlation as suggested in Supplementary Fig. 4). This result demonstrates that machine learning with big data can capture general features and lead to accurate predictions even if uncertainties and noisy components exist in the data source. As shown in Supplementary Fig. 10b, c, the predictions for the $O_2$ landscape remain unchanged when using time series data with either 50-Myr or 100-Myr resolutions.

We have tested our SVR modeling of $O_2$ contents using the earlier database of Keller and Schoene. As shown in Supplementary Fig. 10d, the overall $O_2$ pattern is mostly a close match with predictions using the updated database assembled for this study, suggesting that $O_2$ modeling using the global mafic rock database is robust. Moreover, we have used the CLR-transformation to address the problem of spurious correlations faced in geochemical compositional data analysis[75]. It is beyond the scope of this study to interpret the CLR-transformed concentration values for each element, but the results in Supplementary Fig. 11 suggest no significant differences in both the PCA1 pattern and SVR modeling of the $O_2$ curves whether using raw geochemical data or the CLR-transformed dataset.

We tested additional schemes for reconstructing atmospheric $O_2$ history using different machine learning methods, including ANN and RF. As shown in Supplementary Fig. 12, the resulting first-order pattern and second-order variations for atmospheric $O_2$ levels discussed above remain unchanged. The results suggest that reconstructing atmospheric oxygenation history using machine learning with mafic igneous geochemistry data is robust—independent of machine learning algorithms (e.g., SVR, ANN, and RF). Notably, the RF approach[76] provides an assessment of feature importance for all 44 elements in terms of how each composition variable influences the final model for atmospheric oxygen levels (see Supplementary Fig. 13). In comparison to SVR, the RF method as applied in our study requires significantly more effort for parameter optimization and data training, while the RF and ANN methods provide relatively lower prediction accuracy. More importantly, the results from SVR modeling show much more consistency with $O_2$ levels and variability suggested previously by sedimentary proxies.

## Data availability

Global mafic igneous composition data were assessed through the EarthChem data repository (http://portal.earthchem.org/). All compiled and generated data sources in this study are available at Github repository (https://github.com/myscren/deeptimeML) and have been published on Zenodo (https://doi.org/10.5281/zenodo.7042193).

## Code availability

All computational source codes used in this paper are available on Github repository (https://github.com/myscren/deeptimeML) and have been published on Zenodo (https://doi.org/10.5281/zenodo.7042193).

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

## Acknowledgements

The authors thank Dr. K. Chen and Dr. Cin-Ty A. Lee for their comments that improved this paper. This research was supported by National Natural Science Foundation of China (Grant Nos. 41972305 and 42050103). Q.M.C. received financial support from UNESCO Chair Program in Deep-time Digital Earth and Mineral Resources. G.X.C. received financial support from the MOST Special Fund (No. MSFGPMR2022-3) from State Key Laboratory of Geological Processes and Mineral Resources. Funding was provided to T.W.L. through the NASA Astrobiology Institute under Cooperative Agreement No. NNA15BB03A issued through the Science Mission Directorate and the NASA Interdisciplinary Consortia for Astrobiology Research.

## Author contributions

G.X.C. and Q.M.C. conceived the reconstruction of atmospheric oxygenation history using data-driven machine learning algorithms. G.X.C. drafted the manuscript with substantial contributions from T.W.L., Q.M.C., J.S., and F.A. T.W.L. largely deepened the discussions and improved the conclusions. G.X.C. designed and developed the machine learning approach used in this study, and statistic tests were performed by G.X.C., N.H., and M.L.Z. All authors analyzed the results and revised the article.

## Competing interests

The authors declare no competing interests.
