## [Peer Review File · Nature Communications]

Reconstructing Earth's Atmospheric Oxygenation History Using Machine LearningREVIEWER COMMENTS

Reviewer #1 (Remarks to the Author):

This is a well-written manuscript that uses machine learning on a global data set of basalt compositions to understand temporal trends in the record. The resultant chemical vector space is then used to calculate changes in O₂ resulting from changes in reductants, oxidants, and nutrients being delivered via igneous rocks. The computational exercises are able to reproduce the two-step trends previously inferred for atmospheric oxygenation, as well as, second order fluctuations that some previous studies have suggested. The authors emphasize that long term mantle cooling was the driver of changes in basalt chemistry, and thus changes in atmospheric O₂ concentrations (at least partially). I found the paper interesting and at times provocative, but ultimately it was primarily an exercise in correlation.

My expertise lies in igneous petrology, but my understanding of the machine learning approach here is that there are two records: basalt geochemistry and atmospheric O₂ levels. Both of these show two periods of major transition in earth history in the paleoproterozoic (GOE) and the neoproterozoic (NOE). In particular, basalt geochemistry changes from high Mg, Ni, and other compatible elements with low P and incompatible elements in the Archean to the reverse of this in the Phanerozoic. These can be made sense of in terms of oxidant, reductants, and nutrient sources. But since the igneous and O₂ records are in temporal lock step any supervised machine learning exercise is going to be able to reproduce the changes observed. In other words, the function for atmospheric O₂ prediction didn't come out of a known process, but rather an empirical trend seen in both data sets.

Again, my expertise is not in machine learning but as a skeptical petrologist I'd like to see this concern addressed. I think addressing it explicitly would give the broader community more faith in the conclusions.

There are two other main comments:

(1) If the main trends are being driven by secular mantle cooling, then this would be most seen in MORBs, which are generally not preserved in the rock record and are not sampled here. The dataset used in this manuscript are "continental" in character. I understand the argument about evolution of plate tectonics regimes, but arc-related basalts (which can dominate the continental record) aren't the best record of decreasing mantle potential temperature. They can also be affected by assimilation, fractional crystallization, etc which can change concentrations of various elements. Last, SiO₂ isn't the best indicator of whether a rock is a mantle melt. Even a screen for Mg# (i.e., 65-75) before completing the machine learning exercise would be useful.

Just like the sedimentary rock record, the igneous record is incomplete and certain time periods may be represented by one or just a few localities that are not representative of global conditions. Have you

gone back into periods where the second order variations are observed to see what the igneous dataset is actually comprised of? Are they dominated by flood basalts or another specific lithology? In addition, the data sets may be skewed towards over-represented lithologies simply because people have studied and analyzed those rocks in more detail. This makes me wonder about what the second order variations actually mean.

In any case, I think there is merit in this paper because some interesting inferences have been made and, for better or worse, similar machine learning exercises will continue in the future with their rapid gain in popularity. However, I'm not sure I was convinced of anything. I could be with a little further thoughtful discussion as described above.

Reviewer #2 (Remarks to the Author):

Dear Guoxiong Chen and authors,

I find your study very interesting and expect it to have broad impact. As stated in your study, the sedimentary record has significant shortcomings with respect to temporal coverage. In my view, sedimentary evidence is easily over-interpreted, where, for example, the interpretations of sedimentary geochemical proxies from limited geographical and time intervals are extrapolated to global atmospheric and redox states even in cases where the evidence is more representative of localized, not global, conditions.

The extensive geographical and temporal coverage of your approach, using machine learning on big data from mafic rock geochemistry, gives a way forwards. That said, the discussion about the pros and cons of your approach is thoughtful, and appreciated.

This study's conclusion that mantle cooling controls melting, therefore controls chemical fractionation of mantle melt, therefore controls atmospheric O₂ through time... all seems robust. The resulting Earth atmospheric O₂ history is much more detailed and comprehensive than shown before - this is a major contribution. However, I do have a few suggestions for your consideration that I hope can help improve this study and increase its impact.

1) Please show the temporal coverage of the data set. The geographical coverage of the data is shown in Figure S1. It would also be important to show the temporal coverage of the data, for example in a plot with the number of samples on the y-axis, and sample age on the x-axis.

2) It could help to increase the reach of your paper by adding a quick explanation of 'compatible' and 'incompatible' elements for the general reader.

3) The recorded loss of sulfur mass independent fractionation (S-MIF) signals in rocks (such as $\Delta^{33}\text{S}$ and $\Delta^{36}\text{S}$) has been considered 'smoking gun' evidence for oxygenation through the GOE, ca. 2.5-2.0 Ga (e.g., Farquhar et al., 2000; Kump, 2008). In particular, $\Delta^{33}\text{S}$ signals going below 0.3‰ after 2.3 Ga has been considered a marker of the GOE itself (Bekker et al., 2004; Luo et al., 2016), indicating that atmospheric oxygen levels were above the threshold of S-MIF ($>10^{-5}$ PAL $p\text{O}_2$). Even this year, there has been a new high profile paper, in Nature, further interpreting S-MIF signals for a delay to atmospheric oxygenation, pushing permanent oxygenation to around 2.2 Ga (Poulton et al., 2021). The present study seems at odds with such a conclusion, and this is important. Therefore, it would seem that approaches such as that of the present study offer a sorely needed perspective on Earth's oxygenation, helping our understanding where the interpretation of S-MIF has been insufficient. However, this is not discussed in the present paper. Although it might be 'kicking a hornets' nest' to discuss, considering the importance that the sulfur isotope record has had, for many years, for the interpretation of the GOE, it seems important to make a comparison between the new model results in this paper with the S-MIF record.

For example, the increase in the amplitude of $\Delta^{33}\text{S}$ in the run up to the GOE has not been well explained in the published literature so far. Sulfur isotope $\Delta^{33}\text{S}$ signals reach their maximum at 2.5 Ga, coincident with the 'Archaean Oxygenation Event' revealed and discussed in this new study. Conventional interpretation of the high $\Delta^{33}\text{S}$ at 2.5 Ga would have $p\text{O}_2$ at less than 10^{-5} PAL at that time, while this new study suggests $p\text{O}_2$ was already above this threshold just before 2.5 Ga. Then, low level $\Delta^{33}\text{S}$ signals occurring from ca. 2.45 to 2.2 Ga have been variably interpreted as sedimentary recycling of S-MIF (a la Reinhard et al., 2013), or oscillations of $p\text{O}_2$ (a la Poulton et al., 2021). The present study does not seem to support $p\text{O}_2$ oscillations that would generate S-MIF through the 2.45 to 2.2 Ga time interval, for example. Or perhaps the possibility of $p\text{O}_2$ oscillations through the GOE is still left open in your study... however, this possibility is not discussed. Even if it is not conclusive, a discussion and comparison between the early O_2 history suggested by your work versus what has been interpreted from the $\Delta^{33}\text{S}$ record would be very useful. As it stands, the absence of such discussion seems conspicuous.

Furthermore, the study by Stueken et al., 2012, using a large compilation of sulfide concentration data from sedimentary shales, suggests oxidative weathering of sulfur on land before 2.5 Ga, a timeline of early O_2 rise that supports your result of an 'Archaean Oxidation Event' before the GOE.

I provide some comments on details and include references below.

Regards,

Bryan Killingsworth

USGS

DETAILS:

L87: typo on "dismal".. should be dismissal?

L146: "overshot" ... please change to "overshoot"

L155: "the Snowball Earth" maybe change to "Snowball Earth events"

L184: "Moreover, the coincidence of the two patterns around 0.7 Ga has not been reported previously."
Needs more clarity about what the "two patterns" are, here. Move it to the end of the paragraph?

L222: typo? "atmospheric variation" maybe should be "atmospheric oxygenation" ?

L223: "Thus, the rate of O2 change in the atmosphere can be formulated as:16,62

$$(dm_a)/dt=J_{in}^{(O_2)}-J_{out}^{(O_2)}, (1)$$

where t represents time, m_a represents the mass of O2 in the atmosphere, $J_{in}^{(O_2)}$ represents the photosynthetic production rate of O2, and $J_{out}^{(O_2)}$ is the consumption rate of O2 through reductants derived from Earth's interior."

...what about the O2 outflux via respiration?

...and is the photosynthetic production rate gross production or net production?

L347: ...so the model doesn't capture the "oxygen overshoot"... alternatively, perhaps the "oxygen overshoot" itself is not robust?

L348: change "overshot" to "overshoot"

L364: "oxygenated atmospheric" ... maybe change to "atmospheric oxygenation"

L463: "Also, we have conducted a SVM machine learning prediction using only well-known reductive elements (FeOT, Ni, S) and nutrient elements (P2O5), but the result (as shown in Fig. S7) did not match well with the previous classic model of atmospheric O2 level, especially for NOE."

...caption on S7 does not list S.

REFERENCES:

Farquhar, James, Huiming Bao, and Mark Thiemens. "Atmospheric influence of Earth's earliest sulfur cycle." *Science* 289.5480 (2000): 756-758.

Kump, Lee R. "The rise of atmospheric oxygen." *Nature* 451.7176 (2008): 277-278.

Bekker, A., et al. "Dating the rise of atmospheric oxygen." *Nature* 427.6970 (2004): 117-120.

Luo, Genming, et al. "Rapid oxygenation of Earth's atmosphere 2.33 billion years ago." *Science Advances* 2.5 (2016): e1600134.

Poulton, Simon W., et al. "A 200-million-year delay in permanent atmospheric oxygenation." *Nature* 592.7853 (2021): 232-236.

Reinhard, Christopher T., Noah J. Planavsky, and Timothy W. Lyons. "Long-term sedimentary recycling of rare sulphur isotope anomalies." *Nature* 497.7447 (2013): 100-103.

Stüeken, Eva E., David C. Catling, and Roger Buick. "Contributions to late Archaean sulphur cycling by life on land." *Nature Geoscience* 5.10 (2012): 722-725.

Reviewer #3 (Remarks to the Author):

This is a good article. Authors use ML as a tool to discover broad trends in deep time data. The work supports the conclusions.

A few comments for improvement:

1. Please add more details on how you used SVM for supervised learning. How did you construct and divide training, validation, and test data?
2. Please report your error rate in addition to accuracy (which you have done).
3. Also mention whether there is a possibility of combining any analytical equation with ML algorithms for this study.

Response to reviewer #1:

- 1. Q: My expertise lies in igneous petrology, but my understanding of the machine learning approach here is that there are two records: basalt geochemistry and atmospheric O₂ levels. Both of these show two periods of major transition in earth history in the paleoproterozoic (GOE) and the neoproterozoic (NOE). In particular, basalt geochemistry changes from high Mg, Ni, and other compatible elements with low P and incompatible elements in the Archean to the reverse of this in the Phanerozoic. These can be made sense of in terms of oxidant, reductants, and nutrient sources. But since the igneous and O₂ records are in temporal lock step any supervised machine learning exercise is going to be able to reproduce the changes observed. In other words, the function for atmospheric O₂ prediction didn't come out of a known process, but rather an empirical trend seen in both data sets.*

R: We can clarify our approach to atmospheric O₂ prediction using big data approaches, but without known processes or mathematical model, in terms of data-driven discovery and specifically the fourth paradigm of Hey et al (2009). Researchers often view the world in terms of causality, which is important, while ignoring or downplaying the value of correlation. In fact, causality follows correlation, while correlation does not necessarily expose causality. Our main goal, as an important step forward, was to reveal first the correlations among massive data sets and from that platform begin to explore whether causation exists in these correlations. Because data can be diverse, high-dimensional, and noisy (like the igneous geochemistry data spanning 45 elements), machine learning is needed to find the hidden correlations and patterns that escape human intuition. At the same time, because applications of artificial intelligence in the Earth sciences have not yet reached the level of cognitive intelligence, we emphasize the need to blend human learning and artificial intelligence when searching for the causation that underlies correlation. In the sections 2.2 and 2.3, we explain that we can use mafic compositions to reconstruct atmospheric oxygen levels because the former can modulate the balance of early O₂ sources and sinks.

In the present study, the unsupervised machine learning of global igneous geochemical composition data revealed three major stages of its secular evolution, which correlate well with two-step increases in atmospheric oxygen levels over time. This agreement of completely independent data/approaches is remarkable and suggests links between the two-step rise and perhaps second-order fluctuations in atmospheric O₂ level and deep Earth geochemical process through mantle cooling. In addition, the observed relationships raise previously unknown questions about correlation versus causation and specifically whether global igneous geochemical time series data can indeed quantitatively predict atmospheric O₂ levels over deep time. Further, by using supervised machine learning, we produced a high-resolution temporal pattern of variation in atmospheric oxygen, which successfully and independently reproduces the classic two-step model but also with more detailed fluctuations. These second-order variations are not well known through independent approaches — but are suggested. The agreement we observe is a vital

indication that such fluctuations might be real, with important implications for the history of the biosphere — patterns and mechanisms — while also motivating future research. The main benefit of using machine learning is that we did not need a specific, comprehensive mathematical model, which would be difficult to construct given the present state of knowledge about the patterns and wide-ranging drivers of varying oxygen levels. By combining machine and human learning, we offer novel mechanistic windows to possible drivers of dynamic Precambrian oxygen levels as related to igneous processes, while at the same revealing possible second-order oxygen variability.

Tansley, S., & Tolle, K. M. (2009). The fourth paradigm: data-intensive scientific discovery (Vol. 1). A. J. Hey (Ed.). Redmond, WA: Microsoft research.

- 2. Q: If the main trends are being driven by secular mantle cooling, then this would be most seen in MORBs, which are generally not preserved in the rock record and are not sampled here. The dataset used in this manuscript are “continental” in character. I understand the argument about evolution of plate tectonics regimes, but arc-related basalts (which can dominate the continental record) aren't the best record of decreasing mantle potential temperature. They can also be affected by assimilation, fractional crystallization, etc which can change concentrations of various elements. Last, SiO₂ isn't the best indicator of whether a rock is a mantle melt. Even a screen for Mg# (i.e., 65-75) before completing the machine learning exercise would be useful.*

R: Basaltic magmatism is mostly generated in three main tectonic settings: mantle plume, mid-ocean ridge, and subduction zone (arc). We understand your major concern on whether the igneous geochemistry trends from continental record reflect decreases in mantle partial melting through time as a consequence of secular mantle cooling. Keller and Schoene (2012) have given a specific answer of this question in their Nature paper. We appended key information below, and more the details and tests can be found in the METHODS of Keller and Schoene (2012).

“Assuming our data set provides a representative sample of the crust, four candidate mechanisms could explain the observed trends (Fig. 1) in compatible and incompatible element composition of mafic rocks: (1) similar degree melting from a mantle that is more enriched now than in the past; (2) higher degrees of crustal contamination of otherwise identical basalts in the present; (3) higher degrees of fractionation in the present; or (4) higher mantle melt fraction in the past, due to either higher temperature or higher water content.”

“A modern-day mantle that is more enriched now than in the Archaean is inconsistent with well established evidence from Nd, Sr and Hf isotope studies of mid-ocean ridge basalts, eliminating mechanism (1). Increasing crustal contamination (2) should be reflected in ratios such as Nb/U and Pb/ Ce, which are instead relatively constant through time. If increasing magma fractionation (3) were used to account for the entire observed trend, this would imply that modern basalts are on average created through fractionation

of very-high-MgO magmas such as komatiites, which is clearly not the case.”

“Given that decreasing average mantle melt fraction through time (4) is then the most compatible interpretation of the data, we are faced with two options: secular cooling or mantle dehydration. Evidence for a more hydrous Archaean mantle is equivocal, and it has even been argued that the potentially undercompensated flux of water into the mantle at subduction zones suggests that the hydration state of the mantle is increasing through time, not decreasing. Secular cooling, on the other hand, presents no such caveats and is widely expected on the basis of physical constraints.”

We understand your good suggestion for screening igneous samples by Mg# (i.e., 65-75), but this criterion is unfortunately too restrictive and could remove most of samples in our dataset (only <2000 samples left in the test), because few samples have reported data for both FeO_T and MgO concentrations. In our updated dataset, the basaltic composition was prepared using 43–53 wt% SiO₂ with total alkali (K₂O + Na₂O) <5 wt%.

3. *Q: Just like the sedimentary rock record, the igneous record is incomplete and certain time periods may be represented by one or just a few localities that are not representative of global conditions. Have you gone back into periods where the second order variations are observed to see what the igneous dataset is actually comprised of? Are they dominated by flood basalts or another specific lithology? In addition, the data sets may be skewed towards over-represented lithologies simply because people have studied and analyzed those rocks in more detail. This makes me wonder about what the second order variations actually mean.*

R: We have expanded the global basalt data compilation of Keller and Schoene (2012), from ~26,000s to 52,000s samples using EarthChem, GEOROC, and USGS data repositories. As illustrated in the Method, the compiled data have basalt samples within most of age bins (98% and 88% for 100-Myr and 50-Myr bins, respectively) ranging from 3.8 Ga to the present. The resulting curves for geochemical concentration time series using the updated dataset show temporal variations similar to those of Keller & Schoene (2012) but show more detail with less uncertainty because of the expanded database. We also show that the igneous data time series prepared using 50 Myr and 100 Myr bins show similar second-order variations, suggesting that the second-order variations were not an artifact of the size of age bin. As expected, the ML predictions of O₂ content and the resulting O₂ curves using the above three different time-series are consistent, as shown in supplementary Figs. 5e, 6a, and 6c.

Although preservation and sampling bias are inevitable in all geological records, we use Monte Carlo analysis with weighted bootstrap resampling to calculate the global mean geochemical concentration with time. This method can minimize issues of sampling bias. As illustrated in the METHODS of Keller and Schoene (2012), resampling weights are inversely related to spatiotemporal sample density, and therefore we can achieve a maximally uniform posterior sample density distribution and minimize selection bias

(thus addressing the oversampling issue). Further, this comprehensive sample of continental igneous rocks has been proven sufficiently large and tectonically diverse to capture a global signal in many publications (Cox et al., 2018; Dien et al., 2020; Liu et al.; 2019). Second-order variations of mafic igneous mean concentration were previously reported in Nature Communications to reflect changes in mantle potential temperature and degree of mantle melting, with important implications for mantle dynamics and/or supercontinent formation (Dien et al., 2020). We agree, therefore, the result is a best possible estimate of the average igneous geochemistry of exposed continental crust through time for the present level of data availability.

In addition, machine learning predictions with big data are obtained by complete integration of common features within all geochemical time series data, and the predicted temporal variations can be robust because unrelated anomalous values (outliers) in time series data would not affect the results. As shown in supplementary Fig.5f, there are no differences between the predictions using all 45 elements and when using only the highly correlated elements as suggested by cluster analysis in Fig. 3c (i.e., precluding less correlated elements SiO₂, CaO, MnO, FeO_T, S, Cu, and Sc). We have added related explanations and discussions to the revised manuscript.

Cox, G. M., Lyons, T. W., Mitchell, R. N., Hasterok, D. & Gard, M. Linking the rise of atmospheric oxygen to growth in the continental phosphorus inventory. Earth and Planetary Science Letters 489, 28-36 (2018).

Dien, H. G. E., Doucet, L. S. & Li, Z.-X. Global geochemical fingerprinting of plume intensity suggests coupling with the supercontinent cycle. Nature communications 10, 1-7 (2019).

Liu, H., Sun, W.-d., Zartman, R. & Tang, M. Continuous plate subduction marked by the rise of alkali magmatism 2.1 billion years ago. Nature communications 10, 1-8 (2019).

Response to reviewer #2:

- 1. Q: Please show the temporal coverage of the data set. The geographical coverage of the data is shown in Figure S1. It would also be important to show the temporal coverage of the data, for example in a plot with the number of samples on the y-axis, and sample age on the x-axis.*

R: The temporal coverage of the global data compilation of whole rock igneous compositions has been added in the revised manuscript. We have moved the previous supplementary Figure S1 to Figure 2 in main text.

- 2. Q: It could help to increase the reach of your paper by adding a quick explanation of 'compatible' and 'incompatible' elements for the general reader.*

R: We have added a brief explanation of terminologies, including 'compatible' and

'incompatible' elements at their first appearances, see in Lines 129-131.

- Q:* The recorded loss of sulfur mass independent fractionation (S-MIF) signals in rocks (such as $\Delta^{33}\text{S}$ and $\Delta^{36}\text{S}$) has been considered 'smoking gun' evidence for oxygenation through the GOE, ca. 2.5-2.0 Ga (e.g., Farquhar et al., 2000; Kump, 2008). In particular, $\Delta^{33}\text{S}$ signals going below 0.3‰ after 2.3 Ga has been considered a marker of the GOE itself (Bekker et al., 2004; Luo et al., 2016), indicating that atmospheric oxygen levels were above the threshold of S-MIF ($>10^{-5}$ PAL $p\text{O}_2$). Even this year, there has been a new high profile paper, in Nature, further interpreting S-MIF signals for a delay to atmospheric oxygenation, pushing permanent oxygenation to around 2.2 Ga (Poulton et al., 2021). The present study seems at odds with such a conclusion, and this is important. Therefore, it would seem that approaches such as that of the present study offer a sorely needed perspective on Earth's oxygenation, helping our understanding where the interpretation of S-MIF has been insufficient. However, this is not discussed in the present paper. Although it might be 'kicking a hornets' nest' to discuss, considering the importance that the sulfur isotope record has had, for many years, for the interpretation of the GOE, it seems important to make a comparison between the new model results in this paper with the S-MIF record.

For example, the increase in the amplitude of $\Delta^{33}\text{S}$ in the run up to the GOE has not been well explained in the published literature so far. Sulfur isotope $\Delta^{33}\text{S}$ signals reach their maximum at 2.5 Ga, coincident with the 'Archean Oxygenation Event' revealed and discussed in this new study. Conventional interpretation of the high $\Delta^{33}\text{S}$ at 2.5 Ga would have $p\text{O}_2$ at less than 10^{-5} PAL at that time, while this new study suggests $p\text{O}_2$ was already above this threshold just before 2.5 Ga. Then, low level $\Delta^{33}\text{S}$ signals occurring from ca. 2.45 to 2.2 Ga have been variably interpreted as sedimentary recycling of S-MIF (a la Reinhard et al., 2013), or oscillations of $p\text{O}_2$ (a la Poulton et al., 2021). The present study does not seem to support $p\text{O}_2$ oscillations that would generate S-MIF through the 2.45 to 2.2 Ga time interval, for example. Or perhaps the possibility of $p\text{O}_2$ oscillations through the GOE is still left open in your study... however, this possibility is not discussed. Even if it is not conclusive, a discussion and comparison between the early O_2 history suggested by your work versus what has been interpreted from the $\Delta^{33}\text{S}$ record would be very useful. As it stands, the absence of such discussion seems conspicuous.

Furthermore, the study by Stueken et al., 2012, using a large compilation of sulfide concentration data from sedimentary shales, suggests oxidative weathering of sulfur on land before 2.5 Ga, a timeline of early O_2 rise that supports your result of an 'Archean Oxidation Event' before the GOE.

R: We completely agree that we should do more in this regard and have added text accordingly. Thanks for the suggestions. Indeed, there have been tremendous advances in community's understanding of the protracted fabric of the GOE, as expressed particularly in S-MIF data. We have worked to include essential details in this regard, to the degree that space limitations permit, including the recent Poulton et al. paper and the arguments of Stueken et al. Unfortunately, we do not have space to address all the salient details, nor do we want to overstep the level of interpretation our data and models support. We hope

the reviewer will be satisfied with the balance we have struck.

We added comments on the timing of GOE as linked to our data - Line 302-316: "First, our results (Fig. 5c) indicate Archean Oxidation Events (AOEs) at 3.0 Ga and 2.5 Ga. The 3.0 Ga event coincides generally with the emergence of oxygenic photosynthesis (Planavsky et al., 2014), and the 2.5 Ga event is linked to the whiffs of oxygen discussed previously (Andar et al., 2007). Multiple lines of evidence from sedimentary records are consistent with such events (reviewed in Lyons et al., 2021) and suggest that the GOE was the culmination of an extended period of transient oxidation (Lyons et al., 2014; Stüeken et al., 2012).

In detail, the background atmospheric O₂ level prior to the 2.5 Ga event was less than 10⁻⁷ to 10⁻⁵ PAL, based on the conventional interpretation of sulfur mass independent isotopic fractionations (S-MIF) found in the sedimentary record at that time (in supplementary Fig. 8). Except for a short-term return to low pO₂ after an AOE at 2.5 Ga, however, our data suggest a continuous climb in pO₂ to 10⁻² PAL by 2.1 Ga. Our model predicts that atmospheric O₂ levels exceeded 10⁻⁷ to 10⁻⁵ PAL at ~2.4 Ga, and it does not seem to support the O₂ oscillations suggested by low S-MIF signals occurring from ca. 2.45 to 2.2 Ga (Poulton et al, 2021). However, a higher temporal resolution (< 50 Myr bin) of global igneous geochemistry is required to explore the possibility of these oscillations through the GOE with confidence when using our machine learning model."

Note: given that many dynamic models suggest system stability ~ 10⁻⁷ PAL after the evolution of oxygenic photosynthesis (Pavlov and Kasting, 2002; Goldblatt et al., 2006; Lyon et al., 2021), we labeled O₂ content to 10⁻⁵ - 10⁻⁹ PAL pre 2.5 Ga (in current modeling) instead of 10⁻⁵ - 10⁻⁷ PAL in our previous model. The emerging curve in the revised version has no more than 10⁻⁵ PAL of O₂ at ~2.5 Ga peak.

4. *Q: L87: typo on "dismal".. should be dismissal?*

R: Revised.

5. *Q: L146 and L348: "overshot" ... please change to "overshoot"*

R: Revised.

6. *Q: L155: "the Snowball Earth" maybe change to "Snowball Earth events"*

R: Revised.

7. *Q: L184: "Moreover, the coincidence of the two patterns around 0.7 Ga has not been reported previously." Needs more clarity about what the "two patterns" are, here. Move it to the end of the paragraph?*

R: Revised. We have reworded this sentence in line 187: “the coincidence of the igneous geochemistry transition and atmospheric oxygenation patterns around 0.7 Ga has not been reported previously”

8. **Q:** *L222: typo? "atmospheric variation" maybe should be "atmospheric oxygenation" ?*

R: Revised.

9. **Q:** *"Thus, the rate of O₂ change in the atmosphere can be formulated as: $(dm_a)/dt = J_{in}^{(O_2)} - J_{out}^{(O_2)}$, (1), where t represents time, m_a represents the mass of O₂ in the atmosphere, $J_{in}^{(O_2)}$ represents the photosynthetic production rate of O₂, and $J_{out}^{(O_2)}$ is the consumption rate of O₂ through reductants derived from Earth's interior." ...what about the O₂ outflux via respiration? ...and is the photosynthetic production rate gross production or net production?*

R: We agree that the possibility of aerobic respiration should be considered. We have reworded the text in Line 227-228: “ $J_{in}^{(O_2)}$ represents the net production rate of O₂ (through aerobic photosynthesis and inspiration)”. Quantifying very early aerobic respiration would be extremely difficult given many uncertainties, including the historical records of those processes, and the paucity of tracers that might speak to their ecological significance and net impacts on carbon cycling. This is a topic of our ongoing research. Ours is not a comprehensive model of carbon production, burial, and cycling. Rather, we are assessing what is possible in terms of organic production stimulated by igneous nutrient sources as well as the sinks related to igneous processes. We utilized the Eqs. (1) and (2) with incentive to explain that mafic compositions (nutrients and reductants) can modulate the balance of early O₂ sources ($J_{in}^{(O_2)}$) and sinks ($J_{out}^{(O_2)}$), respectively. This is the underlying mechanism that we need to explain why we can use machine learning with time-series data of mafic igneous geochemical concentration to reconstruct atmospheric oxygen levels. Otherwise, our approach could be a Black Box game because no constitutive relationship when using machine learning algorithm. Our results yield important first- and second-order relationships that are not likely to go away with more comprehensive treatment of carbon-oxygen coupling, although specific magnitudes of oxygen variability would require such efforts. The reality of oxygen variability is likely to have been far more complicated and will require future coupling of diverse data and model types. Those specifics are outside of the goals of the present study. However, our results provide a clear view of the role that igneous processes could have been played. And given the agreement between those igneous-based data/models and independent estimates of varying biospheric redox based on sedimentary geochemical records and biogeochemical models, it is clear to us that the role of igneous processes was important and perhaps among the dominant controls.

10. **Q:** *L347...so the model doesn't capture the "oxygen overshoot"... alternatively, perhaps*

the "oxygen overshoot" itself is not robust?

R: Yes, it is an open question. We have explained the uncertainties of ML modeling in this paper. In line 377-391, “Our model must underestimate O₂ production, as this production in our model is linked only to the flux of nutrient supply from mafic igneous rocks and not recycled from sedimentary sources.” “Our model also underestimates O₂ sinks, as it does not include processes like organic matter remineralization in the water column, weathering of ancient organic in rocks and aerobic inspiration, especially during the Phanerozoic.”

11. **Q:** *L463: "Also, we have conducted a SVM machine learning prediction using only well-known reductive elements (FeOT, Ni, S) and nutrient elements (P2O5), but the result (as shown in Fig. S7) did not match well with the previous classic model of atmospheric O2 level, especially for NOE." ...caption on S7 does not list S.*

R: Revised.

Response to reviewer #3:

1. **Q:** *Please add more details on how you used SVM for supervised learning. How did you construct and divide training, validation, and test data?*

R: Thanks for the thoughtful comment. We have added more details for implementing machine learning in the section of Methods, including unsupervised and supervised learning. We added more details for constructing and dividing training, validation, and test data when using the five-fold cross-validation approach. Please see the details in text: e.g., lines 280-285 and 521-530. Also, we have added details for running our programs and archive the results in this paper in GitHub, <https://github.com/myscren/deeptimeML>.

2. **Q:** *Please report your error rate in addition to accuracy (which you have done).*

R: In the revised manuscript, we have reported both mean accuracy (R² score) and mean-square-error (MSE) in stochastic machine learning for O₂ prediction (lines 286-287).

3. **Q:** *Also mention whether there is a possibility of combining any analytical equation with ML algorithms for this study.*

R: Good suggestions. Although both O₂ sinks and production are related to mantle-derived materials, their relationships with mafic igneous geochemical concentrations are indirect and complex, and it is therefore difficult to build a specific biogeochemical model or quantitative mathematical model for reconstructing atmospheric O₂ content using igneous geochemical concentrations alone. We appreciate the reviewer's concern about using physics-informed or theory-guided machine learning (by combining analytical equation with ML algorithms) to improve the explainability of the current ML model. In future efforts we hope to explore the possibility of using machine learning with igneous geochemistry data in concert with sediment-hosted geochemical proxies to more comprehensively explore the evolution of atmosphere O₂ level. O₂ predictions via machine learning using igneous geochemistry data could be guided or optimized by incorporation of well-established sedimentary biogeochemical proxies and mathematical models that consider carbon sources and sinks quantitatively. Similarly, biogeochemical models/data for oxygen variability and related controls must consider relationships to igneous processes. The overall agreement among independent data and models tells us that all components are important, interrelated, and essential for future efforts at holistic modeling of biospheric evolution.

REVIEWER COMMENTS

Reviewer #2 (Remarks to the Author):

I appreciate that the authors have clearly and thoroughly addressed the concerns I previously raised in my first review.

One small thing to fix:

Line 227-228: " $-J_{in}(O_2)$ represents the net production rate of O₂ (through aerobic photosynthesis and inspiration)"

Please change "inspiration" to "respiration"

Regards,

Bryan Killingsworth

USGS

Reviewer #3 (Remarks to the Author):

Good revisions. Accepted.

Reviewer #4 (Remarks to the Author):

Please enclosed find my partial review of the manuscript.

I was asked to review the manuscript following an initial round of review and so have only examined the revised document.

To begin my review, I first downloaded the data the authors provided on their github website. I generally like to look through the data in these big compilations to make sure it is as described. Based on my initial examinations of the data, I am not confident that the way the data is described in the manuscript matches the actual data. This includes both typos in the data and, more importantly, that not all samples are mafic rocks. Here is what I did before stopping. I did not review the paper carefully (though I read through it) as I was not confident in the validity of the underlying data.

(1) I first looked at MgO content. I'm using this as I have a sense of the MgO content of mafic rocks (as mafic is really a description of mineralogy). When sorting by MgO there is a sample with 67% MgO. The data base does not provide an SiO₂ content (which it should), but this shouldn't be possible based on their filters for silica content. I then tried to look up the sample to figure out what it was. I assumed the sample ID they give in their table is the name in the database (in this case georoc). I note the authors provide no information that I could find on sample type or publication in their compilation. No sample came up in Georoc. I then went to the compilation they got a lot of their samples from (Keller and Schoene, 2012 Nature) and downloaded the supplement to that paper. The names the authors have given are actually the Keller and Schoene specific IDs that differ from the original traceable ID in the georoc database. Keller and Schoene give both (i.e., their ID and the original ID). But, as it stands (as far as I can tell), in this paper in order for a reader to figure out what a sample actually is, they will need to first go to another paper's supplement, search for the name, and then find the real sample name that was in the actual online database.

(2) Beyond the challenges they have set up in terms of validating their data, I then actually looked at the high MgO (67%) point in Georoc. It turns out to be a typo — this typo is actually in the Keller and Schoene database (i.e., someone else made it and it carries over to here), but it immediately made me worried. How many other typos are there in which the database was not downloaded correctly?

(3) I chalked up the upper one to bad luck (i.e., one typo out of fifty thousand probably doesn't matter). I then looked at a high MgO rock from the Archean to see what it was. It was a komatiite, which is a type of ultramafic rock (not mafic as they say their database contains). This confirmed a concern I had in looking at the data that their description of what they did (mafic rocks only) is not correct and they have other things in here. Now maybe it is OK to include ultramafics (and the komatiites are a signal feature of the Archean), but my reading of the paper did not make this clear. Did the authors include peridotites for example? Or cumulates?

(4) As said, komatiites are expected to be found in the Archean (i.e., I wasn't surprised). So I looked at a high MgO rock from the Phanerozoic (KV52924 in their database, real name ID is PBN 86-24). This is

described in Georoc as a lherzolite which is an ultramafic rock, not mafic (so same issue as above). I went to the actual publication to see what it was. It is a xenolith — i.e. it was derived by transporting rock fragments from depth (260 million years old) in a recent eruption (50,000 years). Now maybe it is fine to include peridotite xenoliths, but these do not represent simple igneous rocks forming on or in Earths' surface.

(5) I then looked at the low MgO rocks (again I'm focusing on the MgO ranges as very low and very high MgO is generally a sign that it is not mafic as it indicates a different mineralogy). I picked out an Archean rock that I could find in this paper that was from Keller and Schoene with an MgO of 0.4% (KV19324 in their database; K3/53884 in Georoc) and got the original paper it is from. The data point is in a table in the original publication under 'altered and sulphuric samples', and it has 16.2% sulfur. The geologic area is described by the original authors as follows: "The footwall volcanic sequence was exposed to alteration processes that have largely destroyed the primary mineralogy and affected their chemical composition." It appears to be a volcanogenic massive sulphide deposit. VMS deposits are important, but they are not mafic rocks.

At this point I stopped looking at the database.

I have now looked at the dataset they have provided on two instances. In the first, the description of the data did not match the actual data and they removed ~5000-6000 points when they uploaded the new table. So a ten percent change. The editor noted this was an honest mistake and that is fine with me.

Regardless, the next time I looked at it with them having fixed the table, I was able to, in 30 minutes, find typos, rock types that though igneous, are not what I personally (and I think most people) would call mafic, and the inclusion of samples that are not igneous (i.e., sulfide deposits). Additionally, the dataset is very difficult to check as the actual sample names from the original database are not given, making it a chore for someone to actually figure out the various data points.

Now, the argument may be that what I have found is a minority of the data and that with ~47,000 data points these outliers, typos, and incorrect inclusions are averaged out. But I think this is incumbent on the authors to demonstrate this before the data is interpreted. These big data exercises are only as good as the data used and I, at this point, am worried about the data fidelity.

If you asked me for a recommendation, I would recommend rejection until they sort out their dataset. Regardless, at this point, I have lost confidence in the underlying data. As such, I do not see a reason to review how they worked with their data or their interpretations.

Revision Remarks

We are very grateful to the three reviewers for their thoughtful comments and suggestions. We have carefully checked and addressed all the comments in the revision. All the revisions taken in the main text are highlighted in red.

Response to Reviewer #2:

Q1: I appreciate that the authors have clearly and thoroughly addressed the concerns I previously raised in my first review. One small thing to fix: Line 227-228: " $-J_{in}^{(O_2)}$ " represents the net production rate of O₂ (through aerobic photosynthesis and inspiration)". Please change "inspiration" to "respiration"

R: Revised.

Response to Reviewer #3:

Q1: Good revisions. Accepted.

R: Thanks for your agreement.

Response to Reviewer #4:

Sincere thanks for reviewer #4 to find the errors about the dataset. In this revision, we abandoned the previous way of data compilation by combining two old databases from previous publications (Keller and Schoene (2012, Nature) and Dien et al (2019, NC)). Here, we compiled the up-to-date global mafic igneous composition data directly from EarthChem data repository which includes PetDB, GEOROC, NAVDAT, and USGS database simultaneously.

We prepared the current database in the following steps:

- (1) The raw data were downloaded from the EarthChem Portal (<http://portal.earthchem.org/>) on 6th February, 2022, using the following parameters: sample type = igneous rocks (name from EarthChem Categories), chemistry = 43–51% SiO₂, Age = 0-4000Ma, material = whole rock/rock, chemistry: major elements, trace elements, rare earth elements. We found 8031 samples in NAVDAT, 46563 samples in GEOROC, 16831 samples in USGS, 550 samples in EarthChem, totally 71975 samples were found.
- (2) Based on the file downloaded from EarthChem, we selected mafic samples using: MgO < 18%, composition = mafic, rock name = basalt (40547 samples), gabbro (2980), diabase (1588), tholeiite (3084), dolerite (1591), basanite (2399), basanitoid (26), trachybasalt (1329), alkali basalt (211). We only used the element which has more than 10000 samples, and 44 elements were used in this study. This new database contains ~54,000 whole-rock

analyses of major, trace and rare earth elements from the mafic igneous rocks.

- (3) We deleted the outliers in geochemical concentration data of each element by using $\text{mean} \pm 3\delta$ thresholding during the Archean, Proterozoic and Phanerozoic time, respectively (see supplementary Fig.2). Note that most of trace and rare earth elements follow lognormal distribution (supplementary Fig.1) and logarithm transformation was therefore applied to these element data for outlier filtering. This process of sample screening was automatically implemented using the compiled MATLAB code MCBg.m.

Q1: I first looked at MgO content. I'm using this as I have a sense of the MgO content of mafic rocks (as mafic is really a description of mineralogy). When sorting by MgO there is a sample with 67% MgO. The data base does not provide an SiO2 content (which it should), but this shouldn't be possible based on their filters for silica content. I then tried to look up the sample to figure out what it was. I assumed the sample ID they give in their table is the name in the database (in this case georoc). I note the authors provide no information that I could find on sample type or publication in their compilation. No sample came up in Georoc. I then went to the compilation they got a lot of their samples from (Keller and Schoene, 2012 Nature) and downloaded the supplement to that paper. The names the authors have given are actually the Keller and Schoene specific IDs that differ from the original traceable ID in the georoc database. Keller and Schoene give both (i.e., their ID and the original ID). But, as it stands (as far as I can tell), in this paper in order for a reader to figure out what a sample actually is, they will need to first go to another paper's supplement, search for the name, and then find the real sample name that was in the actual online database.

R: Much more information has been provided in the new compiled data file downloaded from the EarthChem Portal, and one can easily find the sample ID, sources, references, material, sample composition, rock name, and many others.

Q2: Beyond the challenges they have set up in terms of validating their data, I then actually looked at the high MgO (67%) point in Georoc. It turns out to be a typo — this typo is actually in the Keller and Schoene database (i.e., someone else made it and it carries over to here), but it immediately made me worried. How many other typos are there in which the database was not downloaded correctly?

R: In the revised work, the mafic rock samples were selected using $\text{MgO} < 18\%$. In order to delete the outliers in the data like the typo mentioned above, we deleted the outliers in geochemical concentration data of each element by using $\text{mean} \pm 3\delta$ thresholding during the Archean, Proterozoic and Phanerozoic time, respectively. As shown in supplementary Fig.2, outliers in data have been deleted automatically using the compiled MATLAB code MCBg.m. For example, after the above outlier filtering, we have selected samples with $\text{Al}_2\text{O}_3 < 25\%$, $\text{FeOT} < 20\%$, and $\text{CaO} < 20\%$.

Q3: I chalked up the upper one to bad luck (i.e., one typo out of fifty thousand probably

doesn't matter). I then looked at a high MgO rock from the Archean to see what it was. It was a komatiite, which is a type of ultramafic rock (not mafic as they say their database contains). This confirmed a concern I had in looking at the data that their description of what they did (mafic rocks only) is not correct and they have other things in here. Now maybe it is OK to include ultramafics (and the komatiites are a signal feature of the Archean), but my reading of the paper did not make this clear. Did the authors including peridotites for example? Or cumulates?

R: The current database did not include ultramafic rocks, we only using mafic rocks such as basalt (40547 samples), gabbro (2980), diabase (1588), tholeiite (3084), dolerite (1591), basanite (2399), basanitoid (26), trachybasalt (1329), alki basalt (211).

Q4: *As said, komatiites are expected to be found in the Archean (i.e., I wasn't surprised). So I looked at a high MgO rock from the Phanerozoic (KV52924 in their database, real name ID is PBN 86-24). This is described in Georoc as a lherzolite which is an ultramafic rock, not mafic (so same issue as above). I went to the actual publication to see what it was. It is a xenolith — i.e. it was derived by transporting rock fragments from depth (260 million years old) in a recent eruption (50,000 years). Now maybe it is fine to include peridotite xenoliths, but these do not represent simple igneous rocks forming on or in Earths' surface.*

R: The current database did not include lherzolite, peridotite, and other ultramafic rocks, we only using mafic rocks such as basalt (40547), gabbro (2980), diabase (1588), tholeiite (3084), dolerite (1591), basanite (2399), basanitoid (26), trachybasalt (1329), and alki basalt (211).

Q5: *I then looked at the low MgO rocks (again I'm focusing on the MgO ranges as very low and very high MgO is generally a sign that it is not mafic as it indicates a different mineralogy). I picked out an Archean rock that I could find in this paper that was from Keller and Schoene with an MgO of 0.4% (KV19324 in their database; K3/53884 in Georoc) and got the original paper it is from. The data point is in a table in the original publication under 'altered and sulphuric samples', and it has 16.2% sulfur. The geologic area is described by the original authors as follows: "The footwall volcanic sequence was exposed to alteration processes that have largely destroyed the primary mineralogy and affected their chemical composition." It appears to be a volcanogenic massive sulphide deposit. VMS deposits are important, but they are not mafic rocks.*

R: Sincere thanks for the reviewer to find the errors about the dataset. No altered mafic rocks were included in the current database.

We did not go back to all publications to check ~54,000 whole-rock analyses of mafic rocks. Nonetheless, we have done our best to check and delete the typos, errors, and outliers in data according to the above data cleaning processes. We hope the reviewer will be satisfied with these efforts.

Reviewers' comments:

Reviewer #4 (Remarks to the Author):

To the editors,

Please find enclosed my review of the manuscript. I recommend rejection. I stopped reviewing the paper once I reached figure 2. At that point, I found clear issues such that I lost confidence in the underlying accuracy of the data. This is the second time I have reviewed the paper. The first time, I noted numerous issues with the data in the paper and the authors changed their data set and filtering techniques. However, very fundamental issues remain. I explain the issues below that led me to the reject recommendation and why I did not review any insights gained based on the data.

If you look at the figure map (2a), the blue points are Archean points. In looking at this figure briefly, I noticed something odd. There are blue points in the middle of various oceans on young oceanic crust. This is impossible as Archean rocks are found on continental crust, not oceanic crust. Just to make sure I wasn't confused, I tracked down a few of these points from the middle of the Pacific and where the Galapagos and Azores are. They are all mistakes (and go back to Georoc) in which the age must have been typed in wrong. Most are hot spots (which is why they stand out). Regardless, it is not acceptable for the figure to be published as is given this glaring error.

This of course worried me immediately and goes back to my original concerns from my prior review about whether the authors were looking at the data being used thoughtfully such that blatant errors were not included. I believe that any Earth scientist looking at this map will note the issue as they violate our basic understanding on the preservation of Archean crust. What I do not know is how many data points on the continents themselves are wrong? Perhaps there are few, but this error immediately eroding my confidence in the data used.

I next discuss figure 2b, which is far more concerning. First, the data is obscured by cutting off the histograms such that anything with more than 500 is not shown. Regardless, there are a few curious peaks in this histogram for the Precambrian data — a peak at 1500 Ma and a peak at about 3200 Ma. I have seen many compilations of this sort over the years, none of them have such peaks and there are not a lot of >3 billion year old rocks anyway.

To try to figure out what was going on, I downloaded the Excel data from the github link given in the paper with the title 'New_ign1_mafic_final.xlsx'. Both peaks are associated with a number of rocks having an unknown age but being placed in age as either 'Proterozoic' or 'Archean'. For example, 990 Precambrian data points have a max age of 2500 Ma and min age of 542 Ma such that their age is 1521 Ma — i.e., my understanding is that these samples have no age control other than being labeled Proterozoic and the age used in the analysis here is the average of the max and min (i.e. the Archean age bounds). The same is true for Archean peak — 1377 data points have a min age of 2500 Ma and max age of 3850 such that the average age is 3175 Ma, which is used in the paper. Again, I assume these points have no age control other than being Archean. In the data set there are 4842 rocks of Precambrian age. Thus 49% of the Precambrian data set is of unknown age. I do not believe, given this, that any data analysis looking for time varying trends is useful or accurate. I note, that by cutting off the histogram at 500 for the max, the oddity of these peaks are suppressed.

As was the case when I last reviewed the paper, at this point I lost confidence in the underlying data in terms of quality control and stopped reviewing. To summarize, I arrived at this by looking at the figure they give (figure 2) and noticing very clear oddities in both the locations given and the histogram peaks. Given these issues, I strongly recommend rejection.

Response to Reviewer #4:

Reviewer #4 has provided an exceptional service to us and the broader community through their diligence. We are very grateful. With all due respect to the reviewer, there may have been some confusion about our data processing methods, leading to inaccurate comments about the data compilation. Of course, we take the blame for not being clearer (e.g., data cleaning schemes are implemented in the code but are not clarified in the paper). Therefore, to fully address these points of possible confusion, we have added clarifying details to the paper (Lines 463-482, 582-587, and Fig. 6b) and supplement (supplementary Fig. 3) so that you and other readers will not reach the same unfortunate conclusions.

Q1: If you look at the figure map (2a), the blue points are Archean points. In looking at this figure briefly, I noticed something odd. There are blue points in the middle of various oceans on young oceanic crust. This is impossible as Archean rocks are found on continental crust, not oceanic crust. Just to make sure I wasn't confused, I tracked down a few of these points from the middle of the Pacific and where the Galapagos and Azores are. They are all mistakes (and go back to Georoc) in which the age must have been typed in wrong. Most are hot spots (which is why they stand out). Regardless, it is not acceptable for the figure to be published as is given this glaring error.

This of course worried me immediately and goes back to my original concerns from my prior review about whether the authors were looking at the data being used thoughtfully such that blatant errors were not included. I believe that any Earth scientist looking at this map will note the issue as they violate our basic understanding on the preservation of Archean crust. What I do not know is how many data points on the continents themselves are wrong? Perhaps there are few, but this error immediately eroding my confidence in the data used.

R: Revised. We are extremely grateful to the reviewer for finding the errors nested in Figure 2. Those points in oceanic crust with Archean ages in the raw database were not included in the calculation of time series data, because they did not have max and min ages and were filtered by removing null values of age uncertainty in the code (in line 128, MCbg.m). We eliminated points in Figure 2 with Precambrian ages from oceanic crust in the first and second version of data compilation, but we unfortunately confused this matter in the third version—although those points are few in number and did not lead to changes in temporal variations in mean geochemical concentrations. Again, however, our thanks for catching this, which was possible only through extremely careful review.

Q2: I next discuss figure 2b, which is far more concerning. First, the data is obscured by cutting off the histograms such that anything with more than 500 is not shown. Regardless, there are a few curious peaks in this histogram for the Precambrian

data — a peak at 1500 Ma and a peak at about 3200 Ma. I have seen many compilations of this sort over the years, none of them have such peaks and there are not a lot of >3 billion year old rocks anyway.

To try to figure out what was going on, I downloaded the Excel data from the github link given in the paper with the title ‘New_ign1_mafic_final.xlsx’. Both peaks are associated with a number of rocks having an unknown age but being placed in age as either ‘Proterozoic’ or ‘Archean’. For example, 990 Precambrian data points have a max age of 2500 Ma and min age of 542 Ma such that their age is 1521 Ma — i.e., my understanding is that these samples have no age control other than being labeled Proterozoic and the age used in the analysis here is the average of the max and min (i.e. the Archean age bounds). The same is true for Archean peak — 1377 data points have a min age of 2500 Ma and max age of 3850 such that the average age is 3175 Ma, which is used in the paper. Again, I assume these points have no age control other than being Archean. In the data set there are 4842 rocks of Precambrian age. Thus 49% of the Precambrian data set is of unknown age. I do not believe, given this, that any data analysis looking for time varying trends is useful or accurate. I note, that by cutting off the histogram at 500 for the max, the oddity of these peaks are suppressed.

R: We particularly appreciate the reviewers’ concerns about data points at 1521 Ma and 3175 Ma with large age uncertainties (*max_age* – *min_age*). In fact, we noted this issue previously and addressed it specifically during preparation of the geochemical time series by using the weighted bootstrap sampling method proposed by Keller and Schoene (2012, Nature). We explain further below to hopefully convince you that (1) temporal variations of mean mafic igneous geochemical concentration in our study are real and (2) those data points with large age uncertainties do not lead to significant artifacts in time-varying trends.

(1) In our study, estimates of mean geochemical concentrations of interest (within each age bin) were generated by Monte Carlo analysis with weighted bootstrap resampling. Specifically, this method used sample weights that are assigned to be inversely dependent on spatiotemporal sample density according to the relationship (Keller and Schoene, 2012):

$$w_i \propto 1 / \sum_{j=1}^n \left(\frac{1}{((z_i - z_j)/a)^2 + 1} + \frac{1}{((t_i - t_j)/b)^2 + 1} \right) \quad (1)$$

In accordance with Equation (1), rock samples from periods where relatively more data points were observed (i.e., much crust is preserved) were bootstrap resampled less than those from time periods with fewer data points in the database (i.e., poorly represented crust). Therefore, the age peaks of mafic rock samples (specifically, oversampling at ages like 1521 Ma and 3175 Ma) will not produce significant artifacts in time-varying trends.

In addition, spatiotemporal sample weighting in conjunction with Monte Carlo

sampling of rocks with large reported age uncertainties results in a continuous geochemical record that smooths out stochastic and stepwise transitions (Keller and Schoene, 2012). This is because most resampled data points with assumed ages ($age + p * age\ uncertainty$, as seen in lines 4-12 in the code `mctask.m`) did not fall in the period of interest (age bin = 50 Ma) and were not included in the calculation for averaging. Therefore, the Monte Carlo method can minimize the effects of data points with large age uncertainties (e.g., 1521 Ma and 3175 Ma), and the slopes of any abrupt trends presented herein are therefore minimum estimates. Next, we explain how this method was applied to our data test.

- (2) In order to evaluate the effects of data points with large age uncertainties at 1521 Ma and 3175 Ma when using Monte Carlo methods, we compare the time series signals obtained from filtered and non-filtered dataset. The Figure A1 below shows that the patterns between filtered and non-filtered data have consistent time-varying trends and are not notably different in terms of peak and trough positions. Thus, the difference does not affect our conclusions. Please note that the difference in amplitudes (notably for the Precambrian) is collectively caused by outlier filtering (defined by $mean \pm 3\delta$) and deletion of data points at 1521 Ma and 3175 Ma. In other words, if we did not use the outlier filtering, the only differences in time series signals lie with the amplitudes at periods of 1525 Ma and 3175 Ma. As shown in Fig.A1f, the predicted patterns for O₂ level between filtered and unfiltered data show consistent time-varying trends and do not have notably different peak and trough positions. Although there is a slight difference in the mean amplitude of the O₂ prediction, most of them fall in the uncertainty range informed by the Monte Carlo Simulation.

Figure A1. Time series record of mean geochemical concentration obtained from filtered (data points at 1521 Ma and 3175 Ma deleted, blue line) and non-filtered dataset (data points at 1521 Ma and 3175 Ma were not deleted, red line). Note the very small

differences between the two trends.

- (3) Further, we used a relative age range ($RAR = (MAXAGE - MINAGE)/AGE * 100$) to systematically investigate the influences of age uncertainties in the database on igneous geochemical time series patterns when using the weighted bootstrap sampling method. Only 31% and 57% of the samples recorded in the EarthChem Portal have reported ages with a relative age range of less than 5% and 50%, respectively. We added a rigorous sample screening scheme (data filtered with $RAR < 5\%$ and 50%) to generate a time series record of mean geochemical concentration. Figure A2 below shows that the patterns between RAR filtered and non-RAR filtered data have consistent time varying trends (including peaks and troughs) and do not show notably different peak positions except for the different amplitudes. As shown in Fig.A2f, the predicted patterns for O_2 between filtered and unfiltered data did not show notable differences except for the amplitudes. Importantly, most of these differences in amplitude fall in the uncertainty range informed by Monte Carlo Simulation. Therefore, the above results confirm that we did not produce spurious time-varying trends and thus did not reach erroneous conclusions.

Figure A2. Time series record of mean geochemical concentration (a-e) and O_2 predictions (f) obtained from filtered data with $RAR < 5\%$ (blue line) and 50% (black line), and non-filtered dataset (red line). Note that the RAR of data points at 1521 Ma and 3175 Ma are 128.7 and 42.5, respectively.

In closing, we'd like to emphasize here that our modeling of O_2 content over the past 4.0 Gyrs using the updated database shows a very similar pattern to the first version based only on the Keller and Schoene (2012) database and the second version using the

combined database of Keller and Schoene (2012, NATURE) and Dien et al (2019, NC). Moreover, we have shown the very small differences between non-RAR filtered and RAR filtered data (suggested by reviewer #4) in this response. These results confirm that O₂ modeling using the global mafic rock database is robust. Importantly, the previous results and discussions are not affected. Nonetheless, we respect the reviewer's concerns, and in order to minimize the risk of other readers reaching the same inaccurate conclusion, we have added the O₂ predictions obtained from the filtered data with RAR < 5% and < 50% in the paper (Fig.6b) to clarify the underlying uncertainties. We have clarified all the details in the supplementary information file (Fig.3) and call them out in the paper (Lines 463-482, 582-587, and Fig .6b).

Again, our deep thanks for bringing these problems to our attention. A more rigorous paper has resulted.

REVIEWERS' COMMENTS

Reviewer #5 (Remarks to the Author):

Dear Editor,

I have read the manuscript, previous reviews and the rebuttal letter.

First, I agree with reviewer #4 comments about suspicious age peaks. Those peaks are indeed artefact, and this is something that has been discussed briefly in Dien et al. (2019) and discussed in more detail in a recent paper by Doucet et al. (2022) in a short communication in Precambrian Research: Pitfalls in using the geochronological information from the EarthChem Portal for Precambrian time-series analysis: <https://doi.org/10.1016/j.precamres.2021.106514>.

In this contribution, we showed a significant issue with age in online databases that cannot be ignored, mainly if bootstrapping is applied (we gave several examples).

Fortunately, the authors did use a similar definition of the relative age range or RAR (see page 3 in our article), which helps filter out samples with uncorrected ages.

Therefore, the approach using the bootstrapping method of Keller et al. (2012) and the RAR method to filter data is adequate. As the RAR method is identical to our published article in which we discuss the origin of age artifact and its effect on bootstrapping, I suggest the authors to cite Doucet et al. (2022) page 22 in their revised version and also on the caption of Figure 6, where they described the method.

Also, age data filtering should be briefly described in the introduction as it significantly impacts the number of samples filtered out of the online database (which is, in practice, a significant pitfall of online databases).

In short, the response to reviewer #4 is adequate, and I have no further comments on the manuscript.

Luc Doucet, Perth, the 8th of August 2022

Response to Reviewer #5:

We greatly appreciate the time that reviewers have spent in reviewing this manuscript. Please see below for a detailed point by point response to all comments.

Q1: First, I agree with reviewer #4 comments about suspicious age peaks. Those peaks are indeed artefact, and this is something that has been discussed briefly in Dien et al. (2019) and discussed in more detail in a recent paper by Doucet et al. (2022) in a short communication in Precambrian Research: Pitfalls in using the geochronological information from the EarthChem Portal for Precambrian time-series analysis: <https://doi.org/10.1016/j.precamres.2021.106514>. In this contribution, we showed a significant issue with age in online databases that cannot be ignored, mainly if bootstrapping is applied (we gave several examples). Fortunately, the authors did use a similar definition of the relative age range or RAR (see page 3 in our article), which helps filter out samples with uncorrected ages.

R: Again, our deep thanks to reviewers 4 and 5 for bringing these problems to our attention. A more rigorous paper has resulted, and the important additional efforts should help set a higher standard for future community use of such databases.

Q1: As the RAR method is identical to our published article in which we discuss the origin of age artifact and its effect on bootstrapping, I suggest the authors to cite Doucet et al. (2022) page 22 in their revised version and also on the caption of Figure 6, where they described the method.

R: We have cited Doucet et al. (2022) in the revised manuscript in Line 474 and Figure 6.

Q2: Also, age data filtering should be briefly described in the introduction as it significantly impacts the number of samples filtered out of the online database (which is, in practice, a significant pitfall of online databases).

R: Thank you for your thoughtful suggestions. We have added a note in Lines 101-103: Nonetheless, it should be emphasized that rigorous screening of rock samples (e.g., age data filtering) is essential when applying big data approaches to global geo-databases.